# Catastrophic Failure of LLM Unlearning via Quantization

**Zhiwei Zhang**[1], **Fali Wang**[1], **Xiaomin Li**[2], **Zongyu Wu**[1], **Xianfeng Tang**[3], **Hui Liu**[3],
**Qi He**[3], **Wenpeng Yin**[1], **Suhang Wang**[1] *

[1]The Pennsylvania State University    [2]Harvard University    [3]Amazon
{zbz5349, fqw5095, zzw5373, wenpeng, szw494}@psu.edu
xiaominli@g.harvard.edu, {xianft, liunhu}@amazon.com

## Abstract

Large language models (LLMs) have shown remarkable proficiency in generating text, benefiting from extensive training on vast textual corpora. However, LLMs may also acquire unwanted behaviors from the diverse and sensitive nature of their training data, which can include copyrighted and private content. Machine unlearning has been introduced as a viable solution to remove the influence of such problematic content without the need for costly and time-consuming retraining. This process aims to erase specific knowledge from LLMs while preserving as much model utility as possible. Despite the effectiveness of current unlearning methods, little attention has been given to whether existing unlearning methods for LLMs truly achieve forgetting or merely hide the knowledge, which current unlearning benchmarks fail to detect. This paper reveals that applying quantization to models that have undergone unlearning can restore the "forgotten" information. We conduct comprehensive experiments using various quantization techniques across multiple precision levels to thoroughly evaluate this phenomenon. We find that for unlearning methods with utility constraints, the unlearned model retains an average of 21% of the intended forgotten knowledge in full precision, which significantly increases to 83% after 4-bit quantization. Based on our empirical findings, we provide a theoretical explanation for the observed phenomenon and propose a quantization-robust unlearning strategy aimed at mitigating this intricate issue. Our results highlight a fundamental tension between preserving the utility of the unlearned model and preventing knowledge recovery through quantization, emphasizing the challenge of balancing these two objectives. Altogether, our study underscores a major failure in existing unlearning methods for LLMs, strongly advocating for more comprehensive and robust strategies to ensure authentic unlearning without compromising model utility. Our code is available at: https://github.com/zzwjames/FailureLLMUnlearning.

## 1 Introduction

Large language models (LLMs) have exhibited remarkable abilities in generating human-like text, owing to their training on extensive datasets (Zhao et al., 2023). However, LLMs can also unintentionally learn and reproduce undesirable behaviors from sensitive training data (Liu et al., 2024a; Sun et al., 2024; Li et al., 2024c; 2025). These behaviors include the unauthorized replication of copyrighted content (Li et al., 2024a), the generation of private information such as contact details (Huang et al., 2022; Yan et al., 2024), and offensive or harmful messages (Chao et al., 2023). Such risks present significant ethical and security concerns, complicating the safe and responsible deployment of LLMs in real-world applications (Yao et al., 2023). Furthermore, laws such as the European Union General Data Protection Regulation (GDPR) (Voigt & Von dem Bussche, 2017) have introduced the "Right to be Forgotten", allowing users to request the removal of their personal data from trained models (Xu et al., 2024a).

---

*Corresponding author.

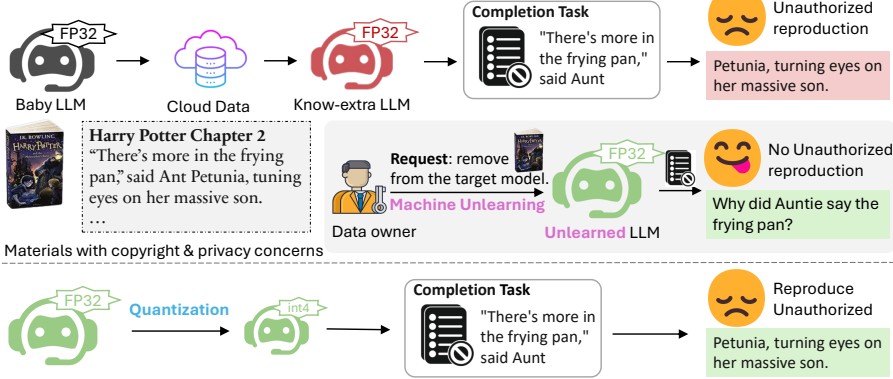

Figure 1: Illustration of the failure of unlearning for LLMs via quantization.

To eliminate the influence of problematic content in the corpora on LLMs, machine unlearning (Liu et al., 2024a; Bourtoule et al., 2021; Liu et al., 2024c; Zhang et al., 2024; Jang et al., 2023; Eldan & Russinovich, 2023; Huang et al., 2024a; Jia et al., 2024; Fan et al., 2024a) has emerged as a promising solution because retraining these models to eliminate undesirable data effects is often impractical due to the costly and prolonged training periods of LLMs. Generally, machine unlearning for LLMs aims to remove the memorization of specific knowledge while maximally preserving utility. Among the advanced unlearning methods, gradient ascent (GA) (Yao et al., 2023) and negative preference optimization (NPO) (Zhang et al., 2024) are the most foundational. GA aims to minimize the likelihood of making correct predictions on a forget dataset by applying gradient ascent to the cross-entropy loss. On the other hand, NPO treats the forget set as negative preference data, adapting the offline DPO (Rafailov et al., 2024) objective to adjust the model to assign a lower likelihood to the forget set. Since GA and NPO are not designed for utility preservation, several regularization techniques (Shi et al., 2024b; Maini et al., 2024) are typically combined with unlearning to preserve utility. For example, given a retain dataset, techniques such as gradient descent on the retain dataset (Zhang et al., 2024; Maini et al., 2024) and minimizing the KL divergence between the unlearned model's and the target model's probability distributions on inputs from the retain dataset (Zhang et al., 2024; Maini et al., 2024) are introduced to enhance the utility of the unlearned model.

Despite their superior unlearning performance, little attention has been given to whether existing unlearning methods for LLMs truly achieve forgetting or merely hide the knowledge, that current unlearning benchmarks fail to detect. In this paper, we discover that *given an unlearned model using existing representative unlearning methods, simply applying quantization can partially or even significantly recover the forgotten knowledge*. Specifically, as shown in Figure 1, given a target model and a forget dataset, we apply unlearning methods to the model to remove knowledge from the forget dataset, resulting in an unlearned model. During testing, the unlearned model demonstrates superior unlearning performance in full precision. However, when we simply apply quantization to the unlearned model, the unlearning performance is compromised. As shown in Table 1, applying the unlearning method GA_KLR on the BOOKS dataset (Shi et al., 2024b) results in the unlearned model retaining only 13% of its original knowledge. However, when the unlearned model undergoes quantization, knowledge retention recovers to approximately 89%. We conduct comprehensive experiments to systematically verify our findings, using various quantization techniques across multiple precisions on different benchmarks, highlighting the generality of the critical issue of knowledge recovery through quantization. We argue that this is a critical issue in real-world applications, as quantization is widely used in the era of LLMs to deploy models in resource-constrained scenarios (Dettmers et al., 2024b; Frantar et al., 2023; Lin et al., 2024; Kim et al., 2024). When fine-tuning a model to forget malicious/private content, it is crucial that the malicious/private content cannot be recovered after the model is quantized. Our key hypothesis is that to achieve unlearning without compromising model utility, existing methods typically adopt a small learning rate and regularization on the retain set, encouraging minimal changes to model weights during unlearning. As a result, the model weights of the target LLM and the unlearned LLM are very close. Hence, quantization is likely to map the weights of the target LLM and the unlearned LLM to the same values, meaning the quantized target LLM and the quantized unlearned LLM have similar weights. Since the quantized target LLM retains most of the forgotten knowledge, the quantized unlearned LLM also recovers that knowledge. We provide theoretical analysis in Section 5.

The catastrophic failure of existing unlearning methods for LLMs motivates us to design frameworks that address the discrepancy between full-precision and quantized models in forgetting knowledge from the forget set. Specifically, based on our analysis, we propose increasing the learning rate for both the forgetting loss and retaining loss. The forgetting loss penalizes the model for retaining information from the forget set, while the retaining loss ensures utility is preserved on the retain dataset. While this approach helps mitigate knowledge recovery through quantization, the aggressive updates driven by the forgetting gradients can cause the model to over-adjust, leading to a decline in overall utility. Additionally, using a large learning rate on the retain dataset can introduce a bias toward the retain data, skewing the model's behavior and degrading its performance on tasks outside the retain dataset. To mitigate the side effects of using a large learning rate for unlearning, we extend the concept of localization-informed unlearning methods (Fan et al., 2024b; Meng et al., 2022; Wu et al., 2023; Wei et al., 2024) by constructing module-level saliency maps to guide the unlearning process, selectively updating only the most influential components related to the data to be forgotten. Our empirical results show that this targeted strategy helps mitigate the risks of aggressive updates, preserves the model's utility, and ensures a more balanced unlearning outcome. However, this framework is highly sensitive to hyperparameter selection, leading to an unstable unlearned model. Our observations inspire future research and advocate for more robust and comprehensive quantization-robust unlearning methods for LLMs.

Our **main contributions** are: (i) We identify a critical issue: applying quantization to an unlearned model can lead to the recovery of forgotten knowledge. We conduct extensive experiments to verify this issue and provide a theoretical explanation. (ii) Our findings represent a fundamental failure in current unlearning methods and introduce a new key objective for LLM unlearning: preventing knowledge recovery through quantization, which also helps to standardize benchmarks for unlearning methods. (iii) We empirically verify our theoretical analysis, make initial efforts to mitigate the identified issue, and conduct comprehensive experiments to inspire future research.

## 2 RELATED WORK

**Machine Unlearning for LLMs.** Machine unlearning, initiated by (Cao & Yang, 2015), adapts trained models to behave as if untrained on specific datasets, crucial for LLMs facing privacy and copyright issues due to indiscriminate web data training. Traditional methods like Newton update removals (Ginart et al., 2019; Guo et al., 2020; Sekhari et al., 2021) are impractical for LLMs due to the complexity of Hessian calculations, prompting newer approaches. These methods split into fine-tuning (Yao et al., 2023; Jang et al., 2023; Chen & Yang, 2023; Maini et al., 2024; Eldan & Russinovich, 2023; Patil et al., 2024; Jia et al., 2024) and in-context unlearning (Pawelczyk et al., 2024; Thaker et al., 2024; Huang et al., 2024a). Fine-tuning utilizes Gradient Ascent (GA) (Yao et al., 2023) to minimize correct predictions on forget datasets by modifying the cross-entropy loss. Negative Preference Optimization (NPO) (Zhang et al., 2024) adjusts offline DPO (Rafailov et al., 2024) to reduce the likelihood of the forget set. To address utility preservation, regularized optimization merges unlearning efficacy with model utility loss, as seen in gradient difference Yao et al. (2023); Maini et al. (2024). In-context methods, using modifications such as labeled demonstrations or post-processing filters, fail to fully address privacy as they require retaining sensitive data (Pawelczyk et al., 2024; Thaker et al., 2024). Huang et al. (2024a) introduces a logit offset method using proxy models, avoiding data retention but not meeting unlearning definitions as they do not match retrained model weights. Despite various studies on machine unlearning for LLMs, our study reveals that existing unlearning methods with regularization struggle with knowledge recovery issues due to minimal weight changes. We propose a simple yet effective solution to mitigate this problem. A more detailed introduction of related work is given in Appendix A.

**Quantization for LLMs.** Quantization reduces LLM storage and computational needs by mapping high-precision parameters to a discrete range without altering the model structure. We focus on post-training quantization (PTQ), which directly quantizes LLMs using calibration datasets to optimize scale factors without retraining. Early PTQ methods typically round weights to the nearest level (RTN) to keep runtimes feasible for large models (Dettmers et al., 2024b; Frantar et al., 2023; Lin et al., 2024; Kim et al., 2024). Advanced PTQ strategies have been developed to enhance performance. For example, GPTQ (Frantar et al., 2023) applies layer-wise quantization updating weights with inverse Hessian information. AWQ (Lin et al., 2024) stores the most impactful weights at high precision and determines scaling with per-channel methods. Despite extensive research, the impact of

quantization on unlearning in LLMs remains largely unexplored, highlighting a significant gap in the field. Recently, Kolbeinsson et al. (2024) studies how interventions such as knowledge editing, model compression, and machine unlearning on LLMs interact. Our research is inherently different from theirs: (i) We conduct extensive experiments to show that quantization could recover the forgotten knowledge of LLM unlearning and provide theoretical understanding to explain the phenomenon; and (ii) We point out the pressing need to develop quantization-robust unlearning and propose a simple and effective framework, which can effectively forget the knowledge in the forget dataset, maintain high utility, and alleviate the recovery issue of quantization.

## 3 PRELIMINARY

In this section, we first revisit machine unlearning and quantization for LLMs in Section 3.1. We then present evidence demonstrating that existing unlearning methods typically employ smaller learning rates and impose constraints on model utility within the retain dataset in Section 3.2. These methods aim to achieve effective unlearning by minimizing weight changes and preserving the model's utility.

### 3.1 MACHINE UNLEARNING AND QUANTIZATION FOR LLMS

**Definition of Machine Unlearning**. Given a pre-trained LLM, consider a dataset $\mathcal{D}_{\text{train}}$ and a model $f_{\text{target}}$ with parameters $\theta$ fine-tuned on $\mathcal{D}_{\text{train}}$, we define the forget set $\mathcal{D}_{\text{forget}} \subset \mathcal{D}_{\text{train}}$ as the specific subset of training data to be forgotten. Machine unlearning aims to eliminate the influence of $\mathcal{D}_{\text{forget}}$ and obtain an unlearned model that behaves like a model $f_{\text{retrain}}$ that was fine-tuned only on the retain set $\mathcal{D}_{\text{retain}} = \mathcal{D}_{\text{train}} \setminus \mathcal{D}_{\text{forget}}$. The unlearning algorithm $\mathcal{U}$ takes $f_{\text{target}}, \mathcal{D}_{\text{forget}}$, and, optionally, $\mathcal{D}_{\text{retain}}$ and outputs an unlearned model $f_{\text{unlearn}} = \mathcal{U}(f_{\text{target}}, \mathcal{D}_{\text{forget}}, \mathcal{D}_{\text{retain}})$. The most commonly used mathematical formulation for optimizing model unlearning is presented below:

$$\min_{\theta} \mathbb{E}_{(x_f, y_f) \in \mathcal{D}_{\text{forget}}} [\mathcal{L}_{\text{forget}}(y_f \mid x_f; \theta)] + \alpha \cdot \mathbb{E}_{(x_r, y_r) \in \mathcal{D}_{\text{retain}}} [\mathcal{L}_{\text{retain}}(y_r \mid x_r; \theta)] \tag{1}$$

where $\mathcal{L}_{\text{forget}}$ is a loss function designed to penalize the model for retaining information about the forget set, $\mathcal{L}_{\text{retain}}$ ensures that utility is preserved on the retain dataset, and $\alpha$ is a regularization parameter used to balance them. Different choices of $\mathcal{L}_{\text{forget}}$ and $\mathcal{L}_{\text{retain}}$ are in the Appendix B.

**Quantization for LLMs**. For quantization, consider a group or block of weights $\mathbf{w}$, the linear operation can be expressed as $y = \mathbf{w}\mathbf{x}$; while the quantized version is denoted as $y = Q(\mathbf{w})\mathbf{x}$, where $Q(\cdot)$ is the quantization function. Specifically, the quantization function is given as (Lin et al., 2024):

$$Q(\mathbf{w}) = \Delta \cdot \text{Round}\left(\frac{\mathbf{w}}{\Delta}\right), \quad \Delta = \frac{\max(|\mathbf{w}|)}{2^{N-1}}, \tag{2}$$

where $N$ is the number of quantization bits, and $\Delta$ is the quantization scale factor (step size) determined by the absolute maximum value of $\mathbf{w}$. Advanced post-training quantization methods, such as AWQ (Lin et al., 2024), adjust the scaling factor for each layer to minimize quantization loss on a calibration dataset. In this paper, we use $Q(f)$ to denote the quantized model $f$. Thus, implementing an unlearning method and then quantizing the unlearned model can be formally written as $Q(\mathcal{U}(f_{\text{target}}, \mathcal{D}_{\text{forget}}, \mathcal{D}_{\text{retain}}))$.

### 3.2 UNLEARNING WITH MINIMAL WEIGHT CHANGE AND UTILITY PRESERVATION

We observe that existing LLM unlearning methods typically use very small learning rates to avoid catastrophic drops in model utility. For example, in three popular benchmarks for LLM unlearning, the MUSE benchmark (Shi et al., 2024b) experiments with a peak learning rate of $1e^{-5}$, the TOFU benchmark (Maini et al., 2024) uses peak learning rates of $1e^{-5}$, $1e^{-6}$, and $5e^{-7}$, and the RWKU benchmark (Jin et al., 2024) explores peak learning rates in the range of $1e^{-8}$ to $1e^{-5}$ via grid search. In contrast, normal training or fine-tuning of LLMs typically use a larger learning rate, e.g., models like Llama3-8B (Dubey et al., 2024) use a peak learning rate of $3e^{-4}$, Llama3-70B uses $1.5e^{-4}$ (Dubey et al., 2024), GPT-3 6.7B uses $1.2e^{-4}$, and GPT-3 13B uses $1e^{-4}$ (Brown, 2020).

Additionally, incorporating a utility preservation constraint on a retain dataset is commonly employed to maintain model utility (Fan et al., 2024b; Shi et al., 2024b; Maini et al., 2024). For instance, in Table 3 of the MUSE benchmark paper (Shi et al., 2024b), gradient ascent with a utility constraint

results in an 18% performance drop, whereas gradient ascent without the constraint results in nearly a 100% drop in utility, despite using a small learning rate.

Existing LLM unlearning methods typically combine the above two strategies, resulting in *minimal weight change* that can "forget" the knowledge in the forget dataset while preserving utility. However, during quantization, there is a significant risk that many model weights of the original model $f$ and its unlearned model $\mathcal{U}(f)$ may map to identical quantized values due to the minimal weight change of unlearning. This overlap in weight representation can cause the quantized unlearned model to closely resemble the quantized target model, which results in the failure of unlearning through quantization.

## 4 CATASTROPHIC FAILURE OF UNLEARNING VIA QUANTIZATION

In this section, we conduct experiments across different precision levels with various quantization techniques to test how quantization affects unlearned models, particularly how quantizing an unlearned model may inadvertently cause the partial recovery of knowledge from the forget dataset. Our investigation includes the following questions: **(Q1)** To what extent does quantization affect the LLM unlearning performance? **(Q2)** What effect does quantization precision (e.g., 4-bit or 8-bit) have on unlearning? **(Q3)** How do different quantization techniques affect unlearning?

### 4.1 EXPERIMENTAL SETUP

**Unlearning Methods.** In our study, we assess six effective unlearning methods for LLMs that incorporate two primary families of unlearning algorithms—**Gradient Ascent (GA)** and **Negative Preference Optimization (NPO)**—along with two strategies for utility preservation. The first family, GA, reduces the likelihood of correct predictions on the forget dataset by applying gradient ascent to the cross-entropy loss (Jang et al., 2023; Ilharco et al.; Yao et al., 2023). The second, NPO, treats the forget set as negative preference data, adapting the offline DPO objective to lower the model's likelihood predictions for this set (Zhang et al., 2024; Rafailov et al., 2024). As GA and NPO do not inherently focus on utility preservation, we employ two regularization strategies to address this gap (Liu et al., 2022; Maini et al., 2024; Zhang et al., 2024): **Gradient Descent on the Retain Set (GDR)** and **KL Divergence Minimization on the Retain Set (KLR)**. The GDR strategy integrates a gradient descent learning objective on the retain set to maintain performance, whereas KLR aims to minimize the KL divergence between the probability distributions of the unlearned and target models during next-token prediction on retain set inputs. By integrating these methods and regularization strategies, we have six distinct approaches for unlearning: GA, GA_GDR, GA_KLR, NPO, NPO_GDR, and NPO_KLR. Further details on these methods are provided in the Appendix B. Experiments and discussion on other unlearning methods (Sheshadri et al., 2024; Li et al., 2024b) are in Appendix J.

**Datasets.** We conduct experiments on MUSE (Shi et al., 2024b), a benchmark for evaluating machine unlearning in language models, using two datasets: **NEWS** and **BOOKS**. The NEWS dataset (Li et al., 2023b) includes recent BBC news articles divided into forget, retain, and holdout sets. The BOOKS dataset (Eldan & Russinovich, 2023) features the Harry Potter series, with original novels as the forget set and related FanWiki materials as the retain set to preserve domain knowledge post-unlearning. Details are in Appendix C.1.

**Metrics.** From the perspective of data owners, expectations for an unlearned model include (1) no verbatim memorization, (2) no knowledge memorization, and (3) no privacy leakage. Conversely, developers prioritize (4) utility preservation on the retain set. Following Shi et al. (2024b), we use four metrics to assess these aspects: (1) **M1. VerMem**, which evaluates verbatim memorization by comparing model continuation outputs to actual tokens using the ROUGE score (VerbMem$(f, \mathcal{D}_{\text{forget}}) = \mathbb{E}_{x \in \mathcal{D}_{\text{forget}}}\text{ROUGE}(f(x_{[1:l]}), x_{[l+1:]})$ where ROUGE (Lin, 2004) assesses similarity between machine output and reference, $x_{[1:l]}$ is the initial $l$ tokens, and $x_{[l+1:]}$ the true continuation.)—lower scores for better unlearning; (2) **M2. KnowMem on $\mathcal{D}_{\text{forget}}$**, which measures knowledge memorization by analyzing responses to tailored knowledge QA pairs (KnowMem$(f, \mathcal{D}_{\text{forget}}) = \mathbb{E}_{(q,a) \in \mathcal{D}_{\text{forget}}}\text{ROUGE}(f(q), a)$), with effectiveness indicated by lower scores; (3) **M3. PrivLeak**, which assesses privacy preservation using the Min-K% method (Shi et al., 2024a), an MIA technique that compares AUC-ROC scores between $\mathcal{D}_{\text{forget}}$ and $\mathcal{D}_{\text{holdout}}$. Then, by comparing the AUC score with that of the retrained model, PrivLeak $= \left(\text{AUC}(f_{\text{unlearn}}) - \text{AUC}(f_{\text{retrain}})\right)/\text{AUC}(f_{\text{unlearn}})$, the optimal scores are near zero, and large deviations suggest poor privacy handling; and (4) **M4.**

Table 1: Comparison of unlearning performance between full-precision and quantized models on NEWS and BOOKS datasets. ↑ implies higher is better, ↓ means lower is better, and → 0 indicates closer to zero is better. Results are presented without percentage symbols, consistent across all tables.

| Method | NEWS | | | | BOOKS | | | |
|---|---|---|---|---|---|---|---|---|
| | M1 ↓ | M2 ↓ | M3 → 0 | M4 ↑ | M1 ↓ | M2 ↓ | M3 → 0 | M4 ↑ |
| Target $f_{target}$ | 58.4 | 63.9 | -99.8 | 55.2 | 99.8 | 59.4 | -57.5 | 66.9 |
| Target $f_{target}$ + Quan.(8 bit) | 40.8 | 66.4 | -99.8 | 54.1 | 99.0 | 45.1 | -57.3 | 65.7 |
| Target $f_{target}$ + Quan.(4 bit) | 34.2 | 54.4 | -99.8 | 48.2 | 85.3 | 36.8 | -60.1 | 50.5 |
| Retrain $f_{retrain}$ | 20.8 | 33.1 | 0.0 | 55.0 | 14.3 | 28.9 | 0.0 | 74.5 |
| Retrain $f_{retrain}$ + Quan.(4 bit) | 18.5 | 36.0 | -2.2 | 46.5 | 13.6 | 24.1 | -3.2 | 62.0 |
| GA | 0.0 | 0.0 | 40.4 | 0.0 | 0.0 | 0.0 | -24.9 | 0.0 |
| GA + Quan.(8 bit) | 0.0 | 0.0 | 39.5 | 0.0 | 0.0 | 0.0 | -25.0 | 0.0 |
| GA + Quan.(4 bit) | 0.0 | 0.0 | 24.5 | 0.0 | 0.0 | 0.0 | -30.1 | 0.0 |
| GA_GDR | 0.0 | 28.9 | 87.1 | 34.2 | 0.0 | 2.9 | -56.5 | 44.2 |
| GA_GDR + Quan.(8 bit) | 0.0 | 26.9 | 93.5 | 33.6 | 0.8 | 3.7 | -52.4 | 43.7 |
| GA_GDR + Quan.(4 bit) | 25.0 | 50.1 | -99.1 | 47.7 | 17.9 | 33.7 | -35.2 | 51.9 |
| GA_KLR | 14.1 | 27.1 | -91.6 | 23.1 | 13.0 | 15.1 | -40.8 | 33.7 |
| GA_KLR + Quan.(8 bit) | 15.3 | 29.0 | -91.7 | 24.5 | 12.4 | 10.1 | -37.9 | 35.1 |
| GA_KLR + Quan.(4 bit) | 33.8 | 50.9 | -99.8 | 45.8 | 75.6 | 34.6 | -60.0 | 51.3 |
| NPO | 0.0 | 0.0 | 14.5 | 0.0 | 0.0 | 0.0 | -22.6 | 0.0 |
| NPO + Quan.(8 bit) | 0.0 | 0.0 | 15.0 | 0.0 | 0.0 | 0.0 | -22.8 | 0.0 |
| NPO + Quan.(4 bit) | 16.2 | 25.4 | -71.6 | 27.9 | 7.0 | 5.3 | -46.9 | 17.8 |
| NPO_GDR | 0.3 | 46.1 | 107.2 | 38.6 | 0.4 | 13.4 | -42.6 | 58.6 |
| NPO_GDR + Quan.(8 bit) | 0.1 | 44.2 | 106.3 | 37.0 | 0.9 | 14.0 | -60.2 | 50.5 |
| NPO_GDR + Quan.(4 bit) | 33.2 | 51.4 | -99.8 | 48.2 | 66.0 | 31.9 | -60.8 | 53.2 |
| NPO_KLR | 16.6 | 36.6 | -94.0 | 33.3 | 12.4 | 13.7 | -40.7 | 35.1 |
| NPO_KLR + Quan.(8 bit) | 17.0 | 37.2 | -93.7 | 29.5 | 11.7 | 11.2 | -37.2 | 23.4 |
| NPO_KLR + Quan.(4 bit) | 34.1 | 53.7 | -99.8 | 48.8 | 70.9 | 34.2 | -60.1 | 50.4 |

**KnowMem on $\mathcal{D}_{\text{retain}}$**, ensuring utility preservation with the same metric (KnowMem$(f, \mathcal{D}_{\text{retain}}) = \mathbb{E}_{(q,a) \in \mathcal{D}_{\text{retain}}}$ROUGE$(f(q), a)$) applied to the retain set, where higher scores indicate better preservation. The first three metrics measure forget performance; the last one is for utility. Additional details are available in the Appendix C.2. More implementation details are in Appendix D.1

**Retrained and Target Models.** Details of the backbone model and the process to obtain the retrained model $f_{\text{retrain}}$ and the target model $f_{\text{target}}$ are provided in Appendix C.3.

## 4.2 IMPACT OF QUANTIZATION ON LLM UNLEARNING

To answer **Q1**, we apply 4-bit quantization using round-to-nearest (RTN) to various unlearned LLMs and compare them to full-precision models. Table 1 presents our main results. From the table, we observe that most quantized models exhibit reduced performance on forgetting metrics (M1 VerMem, M2 KnowMem on $\mathcal{D}_{\text{forget}}$, and M3 PrivLeak), yet show improvement on the utility (M4 KnowMem on $\mathcal{D}_{\text{retain}}$), aligning closer to the performance of $f_{\text{target}}$ without unlearning. This suggests that 4-bit quantization might negatively affect unlearning by inadvertently retaining some knowledge from the forget set while preserving utility. We will explain the cause of this observation in Section 5. An exception is GA, which appears to achieve absolute forgetting even after 4-bit quantization; however, this is misleading as it results from a complete loss of model utility due to a lack of constraints. It is worth noting that for unlearning methods with utility constraints, the unlearned model retains an average of 21% of the intended forgotten knowledge in full precision, which significantly increases to 83% after 4-bit quantization. Additional empirical results on another LLM unlearning benchmark, RWKU (Jin et al., 2024) are provided in Appendix F.

## 4.3 EFFECTS OF QUANTIZATION PRECISION ON UNLEARNING

To address **Q2**, we apply 8-bit quantization to unlearned LLMs. We exclude 2-bit precision models from testing due to their big performance gap compared to full-precision models (Zhu et al., 2023), which contradicts our utility preservation requirements in Section 3.2. The result is also given in Table 1. We observe that models with 8-bit quantization perform similarly to full-precision models due to 8-bit's greater sensitivity to weight changes. This observation suggests that when the precision level drops to a certain point, such as to 4-bit, quantization significantly affects unlearning performance and

Table 2: Results of experiments using various quantization methods on NEWS dataset.

| Method | M1 ↓ | M2 ↓ | M3 → 0 | M4 ↑ |
|---|---|---|---|---|
| Target $f_{\text{target}}$ | 58.4 | 63.9 | -99.8 | 55.2 |
| Target $f_{\text{target}}$ + `Quan.(4 bit)` | 34.2 | 54.4 | -99.8 | 48.2 |
| Retrain $f_{\text{retrain}}$ | 20.8 | 33.1 | 0.0 | 55.0 |
| GA | 0.0 | 0.0 | 40.4 | 0.0 |
| GA + `Quan.(AWQ)` | 0.0 | 0.0 | 38.7 | 0.0 |
| GA + `Quan.(GPTQ)` | 0.0 | 0.0 | 30.0 | 0.0 |
| GA_GDR | 0.0 | 28.9 | 87.1 | 34.2 |
| GA_GDR + `Quan.(AWQ)` | 25.2 | 50.7 | -93.2 | 47.6 |
| GA_GDR + `Quan.(GPTQ)` | 24.8 | 50.4 | -92.9 | 47.7 |
| GA_KLR | 14.1 | 27.1 | -91.6 | 23.1 |
| GA_KLR + `Quan.(AWQ)` | 33.7 | 49.8 | -99.9 | 45.1 |
| GA_KLR + `Quan.(GPTQ)` | 33.2 | 49.3 | -99.8 | 45.3 |
| NPO | 0.0 | 0.0 | 14.5 | 0.0 |
| NPO + `Quan.(AWQ)` | 15.8 | 25.3 | -70.0 | 28.0 |
| NPO + `Quan.(GPTQ)` | 15.9 | 25.3 | -70.2 | 28.0 |
| NPO_GDR | 0.3 | 46.1 | 107.2 | 38.6 |
| NPO_GDR + `Quan.(AWQ)` | 29.4 | 49.6 | -99.8 | 48.1 |
| NPO_GDR + `Quan.(GPTQ)` | 30.1 | 48.9 | -99.8 | 46.6 |
| NPO_KLR | 16.6 | 36.6 | -94.0 | 33.3 |
| NPO_KLR + `Quan.(AWQ)` | 31.6 | 52.0 | -99.8 | 46.7 |
| NPO_KLR + `Quan.(GPTQ)` | 32.8 | 51.1 | -99.8 | 46.6 |

could potentially lead to catastrophic failures. Overall, quantized models with low precision, such as 4-bit, tend to recover knowledge from the forget dataset, highlighting substantial risks of **catastrophic failures of unlearning via quantization**. We will explain the cause of these observations in Section 5. Further analysis and evidence of unlearning failures on the RWKU benchmark (Jin et al., 2024) are detailed in Appendix E and F, respectively.

### 4.4 INFLUENCE OF VARIOUS QUANTIZATION TECHNIQUES ON UNLEARNING

To address **Q3**, we apply two advanced 4-bit quantization methods, GPTQ (Frantar et al., 2023) and AWQ (Lin et al., 2024), which differ from RTN by using calibration datasets, often comprising general corpora such as texts from Wikipedia (Frantar et al., 2023), to minimize quantization errors. We conduct experiments under the same experimental settings as in Section 4.2 and the results on the NEWS dataset are reported in Table 2. We can observe that GPTQ and AWQ perform similarly to RTN. Despite efforts to adjust parameters effectively, the calibration datasets, being general rather than tailored to match the domain of the forget dataset, mean that GPTQ and AWQ are still likely to retain knowledge intended to be forgotten. This underscores the pervasive nature of our identified issue: *irrespective of whether quantization methods utilize calibration datasets, quantized unlearned models continue to suffer from failures of unlearning via quantization.*

## 5 EXPLANATION OF THE FAILURE OF UNLEARNING VIA QUANTIZATION

Our observations in Section 4 have indicated that 4-bit quantized models, regardless of the quantization technique used, exhibit poor unlearning performance when compared to their full-precision models. In contrast, 8-bit quantized models achieve performance metrics similar to those of full-precision models. In this section, we aim to explain these phenomena through a theoretical analysis of the quantization mechanism. We use int-4 and int-8 as examples for illustration.

According to the definition in Equation 2, a weight $w$ within a quantization interval $\mathcal{I}_i$ is mapped to a low-precision quantization index $i = \text{Round}(\frac{w}{\Delta})$ within the range $[-2^{N-1}, 2^{N-1} - 1]$, and to a quantized value $q_i = i\Delta$. All weights within interval $\mathcal{I}_i$ are mapped to the same index $i$ and quantized value $q_i$, as defined by:

$$\mathcal{I}_i = \left[ \left( i - \frac{1}{2} \right) \Delta, \ \left( i + \frac{1}{2} \right) \Delta \right), \tag{3}$$

where $\Delta$ denotes the quantization scale factor, dictating the size of each interval. For example, $\Delta_{\text{int4}} = \frac{\max(|\mathbf{w}|)}{2^4 - 1} = \frac{\max(|\mathbf{w}|)}{8}$, and $\Delta_{\text{int8}} = \frac{\max(|\mathbf{w}|)}{2^8 - 1} = \frac{\max(|\mathbf{w}|)}{128}$. In scenarios where $\max |\mathbf{w}| = 200$,

as depicted in Figure 2, all weights within the interval $[-12.5, 12.5)$ map to $q_0 = 0$ under an int-4 precision format. To differentiate the quantized weights of the original model $f$ from those of the unlearned model $f_{\text{unlearn}}$, the weight changes in $f_{\text{unlearn}}$ must exceed the quantization step size $\Delta$. As discussed in Section 3.2, effective unlearning methods that preserve utility typically have minimal weight changes, resulting in $f_{\text{target}}$ and $f_{\text{unlearn}}$ being highly similar, i.e., $Q(f_{\text{unlearn}}) \approx Q(f_{\text{target}})$. We also know that direct quantization of the original model, i.e., applying $Q(f_{\text{target}})$, generally preserves a significant portion of the model's knowledge (Liu et al., 2024b; Egashira et al., 2024; Hong et al., 2024), as quantization approximates the weights while maintaining the model's structural and functional integrity. The similarity between $Q(f_{\text{unlearn}})$ and $Q(f_{\text{target}})$ indicates that the quantized unlearned model may inadvertently retain knowledge from the forget set, even though the full-precision unlearned model successfully eliminates such information.

Furthermore, the notable disparity in performance between int-4 and int-8 can be attributed to the larger mapping interval $\Delta_{\text{int4}}$ relative to $\Delta_{\text{int8}}$. This significant interval size means that minor weight modifications are less likely to influence the quantized values in 4-bit quantization than in 8-bit. As illustrated in Figure 2, only when weight changes exceed $12.5$, int-4 quantized models will reflect these differences. By contrast, int-8 quantization only needs a small change of $0.78125$ in the raw model in order to result in a change in the quantization, and achieving the necessary changes of $0.78125$ is comparatively easier with int-8 quantization. Thus, int-4 quantized models are more likely to fail in unlearning tasks compared to int-8 models.

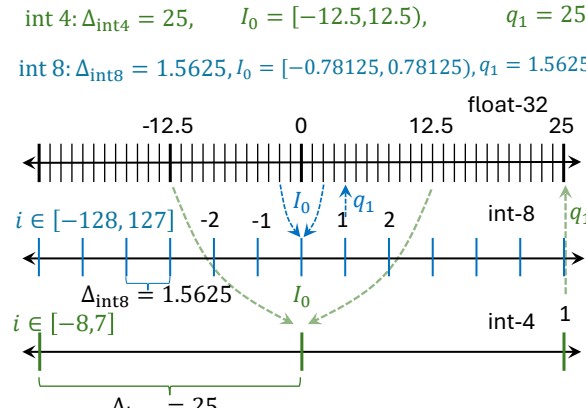

Figure 2: Example of precision loss during model parameter quantization from float-32 to int-4/int-8, with $\max|\mathbf{w}| = 200$. Float values within certain ranges are rounded to the nearest integer.

## 6 QUANTIZATION-ROBUST UNLEARNING

The catastrophic failure underscores the need for effective methods to prevent knowledge recovery while preserving utility. Thus, we propose a tailored strategy based on our theoretical analysis.

### 6.1 PROPOSED FRAMEWORK

We aim for an ideal unlearning method to achieve three key objectives: **(i)** effectively unlearn knowledge from the forget dataset; **(ii)** preserve model utility on the retain dataset; and **(iii)** prevent the recovery of forgotten knowledge through quantization. Based on our theoretical analysis in Sec. 5, the core issue behind the failure of existing unlearning methods in preventing knowledge recovery lies in the fact that effective unlearning seeks minimal weight changes to preserve model utility. This creates a conflict between objectives (ii) and (iii).

One intuitive approach to address the conflict is to increase the learning rate for both $\mathcal{L}_{\text{forget}}$ and $\mathcal{L}_{\text{retain}}$. Intuitively, increasing the learning rate for $\mathcal{L}_{\text{forget}}$ can help achieve objectives (i) and (iii), while the utility constraint imposed by $\mathcal{L}_{\text{retain}}$ on the retain dataset can assist the model in maintaining its performance on that dataset, thus fulfilling objective (ii). However, using a large learning rate to fully fine-tune the model can lead to over-adjustment due to aggressive forgetting gradients, degrading overall utility. Furthermore, applying a large learning rate to the retain dataset may bias the model towards this data, skewing its behavior and further reducing performance on tasks beyond the retain dataset, as demonstrated in Appendix I.

On the other hand, it is acknowledged that large language models may store knowledge in specific neurons (Liu et al., 2024a; Dai et al., 2022), suggesting that unlearning certain knowledge can be achieved by selectively updating model weights, thus minimizing the impact on model utility. Following this idea, we draw on approaches from prior work (Fan et al., 2024b; Meng et al., 2022; Wu

et al., 2023; Wei et al., 2024) and propose constructing a weight saliency map by utilizing the gradient of the loss $\mathcal{L}_{\text{forget}}$ with respect to the model weights on the forget dataset, i.e., $\nabla_{w_i} \mathcal{L}_{\text{forget}}(\theta; \mathcal{D}_{\text{forget}})$. Generally, large magnitude of the gradient, i.e., $|\nabla_{w_i} \mathcal{L}_{\text{forget}}(\theta; \mathcal{D}_{\text{forget}})|$, means the weight $w_i$ is more relevant to the knowledge to be forgotten. We hence choose the weights with large gradients as the saliency weights and update only the salient weights to minimize the potential bias caused by fully fine-tuning with a large learning rate on the retain dataset. In practice, designing a mask for each weight in the era of LLMs is not feasible. Hence, we choose to construct a module-level saliency mask instead. Specifically, we decompose the pre-unlearning model parameters $\theta_o$ into two components: the *salient modules* that will be updated during unlearning and the *intact modules* that will remain unchanged. Specifically, in transformer-based LLMs, the model consists of multiple layers, each containing modules such as multi-head attention mechanisms and feed-forward networks. For each module $i$, let $\theta_i$ denote the parameters associated with that module (e.g., the weights of a specific attention head or feed-forward sub-layer). We compute a saliency score $s_i$ for each module by aggregating the gradients of the forgetting loss with respect to $\theta_i$:

$$s_i = \left\| \nabla_{\theta_i} \mathcal{L}_{\text{forget}}(\theta; \mathcal{D}_{\text{forget}})|_{\theta = \theta_o} \right\|, \tag{4}$$

where $\| \cdot \|$ denotes an appropriate norm (e.g., the Frobenius norm for matrices) that summarizes the gradient magnitudes for module $i$. We then apply a hard thresholding operation to obtain the module-level saliency map $m_M$:

$$m_M[i] = \begin{cases} 1, & \text{if } s_i \geq \gamma, \\ 0, & \text{otherwise}, \end{cases} \tag{5}$$

where $\gamma > 0$ is a threshold value. Hence, those modules with $m_M[i] > 0$ are treated as salient modules to be updated and those with $m_M[i] = 0$ are intact modules. Based on the module-level saliency map $m_M$, we explicitly express the unlearned model parameters $\theta_u$ as:

$$\theta_u = \theta_o + m_M \odot \Delta\theta, \tag{6}$$

where $\Delta\theta$ represents the parameter updates computed during unlearning, $m_M \odot \Delta\theta$ denotes the module-wise multiplication of the saliency mask $m_M$ with the updates $\Delta\theta$. The mask $m_M[i]$ is applied to all parameters associated with module $i$. This formulation implies that during unlearning, we update only the salient modules identified by the module-level saliency map, leaving the rest of the network unchanged. By focusing on module-level saliency, we direct the unlearning process to the most influential parts of the network with respect to the forgetting dataset $\mathcal{D}_{\text{forget}}$. It mitigates the risk of bias toward the retain dataset that could arise from full fine-tuning with a large learning rate. We name this approach Saliency-Based Unlearning with a Large Learning Rate (**SURE**).

## 6.2 EXPERIMENTS

**Experimental Setup.** To thoroughly evaluate our method, following (Jin et al., 2024), we not only test the model's utility on the retain dataset but also assess its performance across various capabilities, detailed as follows: (1) **General Ability (Gen)**: We use MMLU (Hendrycks et al., 2021), which contains multiple-choice questions from diverse knowledge domains. We report 5-shot accuracy based on answer perplexity. (2) **Truthfulness (Tru)**: To evaluate whether the model becomes dishonest after unlearning, we use TruthfulQA's MC1 task (Lin et al., 2022), reporting 6-shot accuracy scores. (3) **Factuality (Fac)**: Since unlearning negates original knowledge, we assess factuality using TriviaQA (Joshi et al., 2017) with 6-shot F1 scores. (4) **Fluency (Flu)**: To measure generation quality, we adopt the instructions in AlpacaEval (Li et al., 2023a) and report the weighted average of bi- and tri-gram entropies (Meng et al., 2022; Zhang et al., 2018).

According to our three objectives, we aim for the incorporation of SURE to achieve comparable forgetting performance and model utility in the full-precision model, as compared to methods without using SURE. Additionally, SURE should help improve forgetting performance after quantizing the unlearned model. Thus, in our experiments, we incorporate SURE into various unlearning methods with regularization and compare them to the original unlearning methods. The implementation of the original unlearning methods follows the setup in Appendix D.1. For each method, we evaluate both the forgetting performance and model utility in full precision and in the quantized version.

We conduct a grid search for the learning rate over the values $[5e^{-5}, 1e^{-4}, 2e^{-4}]$, for the regularization weight $\alpha$ over $[1, 20, 100, 300, 400]$, and for the threshold to construct the saliency mask $\gamma$

Table 3: Results of SURE on BOOKS dataset.

| Method | Forget | | | Utility ↑ | | | | |
|---|---|---|---|---|---|---|---|---|
| | M1 ↓ | M2 ↓ | M3 → 0 | M4 | Gen | Tru | Fac | Flu |
| Target $f_{\text{target}}$ | 99.8 | 59.4 | -57.5 | 66.9 | 28.7 | 33.6 | 9.1 | 573.3 |
| GA_GDR | 0.0 | 2.9 | -56.5 | 44.2 | 22.8 | 35.1 | 6.7 | 563.5 |
| GA_GDR + `Quan.(4 bit)` | 17.9 | 33.7 | -35.2 | 51.9 | 21.4 | 32.7 | 6.0 | 553.6 |
| GA_GDR + SURE | 0.0 | 0.3 | -6.4 | 49.3 | 29.2 | 0.2 | 0.0 | 544.9 |
| GA_GDR + SURE + `Quan.(4 bit)` | 0.0 | 4.8 | -6.3 | 46.2 | 30.4 | 0.18 | 0.0 | 524.7 |
| GA_KLR | 23.8 | 25.1 | -54.5 | 51.9 | 26.2 | 35.7 | 6.7 | 572.7 |
| GA_KLR + `Quan.(4 bit)` | 75.6 | 34.6 | -60.0 | 51.3 | 22.6 | 33.4 | 6.2 | 543.2 |
| GA_KLR + SURE | 16.6 | 25.3 | -57.9 | 46.5 | 22.8 | 28.6 | 9.7 | 546.7 |
| GA_KLR + SURE + `Quan.(4 bit)` | 16.4 | 28.4 | -58.6 | 35.5 | 21.0 | 29.8 | 8.3 | 542.1 |
| NPO_GDR | 3.2 | 27.4 | -51.2 | 57.0 | 25.2 | 35.5 | 7.3 | 570.5 |
| NPO_GDR + `Quan.(4 bit)` | 66.0 | 31.9 | -60.8 | 53.2 | 24.8 | 35.7 | 6.7 | 540.5 |
| NPO_GDR + SURE | 0.0 | 31.2 | -48.2 | 46.1 | 25.2 | 39.5 | 7.3 | 505.9 |
| NPO_GDR + SURE + `Quan.(4 bit)` | 0.0 | 24.4 | -48.1 | 43.2 | 25.1 | 37.2 | 6.3 | 497.9 |
| NPO_KLR | 22.6 | 22.7 | -54.9 | 50.9 | 27.5 | 35.0 | 7.2 | 565.9 |
| NPO_KLR + `Quan.(4 bit)` | 70.9 | 34.2 | -60.1 | 50.4 | 27.0 | 34.3 | 6.5 | 545.6 |
| NPO_KLR + SURE | 17.6 | 37.8 | -58.0 | 49.4 | 23.4 | 30.2 | 7.4 | 588.8 |
| NPO_KLR + SURE + `Quan.(4 bit)` | 16.1 | 36.9 | -58.9 | 34.9 | 23.4 | 31.1 | 8.0 | 592.6 |

over [Percentile($s$, 90), Percentile($s$, 95), Percentile($s$, 99)], where Percentile() refers to the specified percentile over the saliency scores in $s$. Other settings are the same as those in Section D.1. More implementation details can be found in Appendix D.2.

**Results of Unlearning.** We report the results of SURE on the BOOKS dataset in Table 3, with additional results on the NEWS dataset provided in Appendix G. As shown in Table 3, we observe: (i) For quantized models, incorporating SURE significantly improves forgetting performance compared to original methods without SURE. (ii) For full-precision models, incorporating SURE into various unlearning approaches typically achieves comparable forgetting performance and model utility to the original methods. Though for the original unlearning method GA_GDR, our SURE leads to a utility drop in terms of factuality and truthfulness, it still achieves good results in terms of general ability and fluency. This verifies the concern of potential bias introduced by a large learning rate on the retain dataset and highlights the trade-off between maintaining model utility and preventing knowledge recovery through quantization during unlearning. Additional experimental results on hyperparameter analysis and ablation study are provided in Appendix H and Appendix I.

# 7 CONCLUSION

This paper identifies a critical issue: applying quantization to models that have undergone unlearning can restore the "forgotten" knowledge. We conduct comprehensive experiments across various quantization techniques and precision levels to thoroughly evaluate this phenomenon. Furthermore, we provide a theoretical explanation for why this issue occurs. Based on these findings, we propose a saliency-based unlearning strategy using a large learning rate to prevent knowledge recovery via quantization while maintaining model utility. Our study highlights a significant drawback in current LLM unlearning methods and points out an overlooked aspect in existing benchmarks. We strongly advocate for more robust strategies to ensure genuine unlearning without sacrificing model utility.

## ACKNOWLEDGMENT

This material is based upon work supported by, or in part by the Army Research Office (ARO) under grant number W911NF-21-10198, the Department of Homeland Security (DHS) under grant number 17STCIN00001-05-00, and Cisco Faculty Research Award. The views and conclusions contained in this material are those of the authors and should not be interpreted as necessarily representing the official policies, either expressed or implied, of the funding agencies.

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

## A    DETAILED RELATED WORK

### A.1    MACHINE UNLEARNING FOR LLMS

Machine unlearning, initiated by (Cao & Yang, 2015), adapts trained models to behave as if they had never been trained on specific datasets. This is crucial for LLMs, which often face privacy and copyright issues due to training on extensive, indiscriminately collected web data (Wang et al., 2024c). Traditional methods (Ginart et al., 2019; Guo et al., 2020; Sekhari et al., 2021) involve Newton update removals, which are impractical for LLMs due to the computational complexity of Hessian calculations. As a result, newer approaches for LLMs (Jang et al., 2023; Chen & Yang, 2023; Yao et al., 2023; Eldan & Russinovich, 2023; Zhang et al., 2024; Wang et al., 2024a; Huang et al., 2024a; Jia et al., 2024) have emerged. These methods are categorized into fine-tuning based (Yao et al., 2023; Jang et al., 2023; Chen & Yang, 2023; Maini et al., 2024; Eldan & Russinovich, 2023; Patil et al., 2024; Jia et al., 2024) and in-context based unlearning (Pawelczyk et al., 2024; Thaker et al., 2024; Huang et al., 2024a). Fine-tuning-based methods utilize Gradient Ascent (GA) (Yao et al., 2023; Jang et al., 2023; Chen & Yang, 2023; Maini et al., 2024; Jia et al., 2024) to reduce correct predictions on forget datasets by altering the cross-entropy loss. Negative Preference Optimization (NPO) (Zhang et al., 2024) adapts the offline DPO (Rafailov et al., 2024) to lower likelihoods on the forget set. Techniques also include relabeling answers with non-sensitive equivalents to enhance responses Eldan & Russinovich (2023); Patil et al. (2024). To address utility preservation, regularized optimization objectives integrate unlearning efficacy loss with model utility loss, as seen in approaches such as gradient difference Yao et al. (2023); Maini et al. (2024). Moreover, localization-informed methods, focusing on neuron editing (Wu et al., 2023; Yu et al., 2023), remain underexplored for LLMs and large forget datasets and are not discussed in this paper. In-context methods, using modifications such as labeled demonstrations or post-processing filters, fail to fully address privacy as they require retaining sensitive data (Pawelczyk et al., 2024; Thaker et al., 2024). Huang et al. (2024a) introduced a logit offset method that estimates adjustments for unlearning using proxy models, eliminating the need to retain sensitive data. However, these methods do not meet the strict definition of unlearning as they do not ensure model weights match those of a retrained model. True unlearning for LLMs primarily relies on fine-tuned methods, yet existing studies overlook the unlearning performance of quantized models. Our research is the first to thoroughly examine LLM quantization's impact on unlearning. In contrast, the closest study (Kolbeinsson et al., 2024) focuses solely on quantization's effect on unlearning but overlooks utility preservation, leading to conclusions that diverge from ours.

### A.2    QUANTIZATION FOR LLMS

Quantization reduces the storage and computational demands of LLMs by mapping high-precision parameters to a discrete range without changing the model structure. Existing methods for LLMs can be generally categorized into Quantization-Aware Training (QAT) and Post-Training Quantization (PTQ) Wang et al. (2024b). QAT, such as QLoRA (Dettmers et al., 2024a) and IR-QLoRA (Qin et al., 2024), retrains LLMs at low-bit precision but is computationally intensive. PTQ directly quantizes LLMs using calibration datasets to optimize scale factors without the need for retraining. Early PTQ approaches typically round weights to the nearest (RTN) quantization level to maintain feasible runtimes for large models (Dettmers et al., 2024b; Frantar et al., 2023; Lin et al., 2024; Kim et al., 2024). To improve the performance of quantization, more advanced PTQ strategies are developed. For example, SpQR (Dettmers et al., 2024b) uses the L2 error between original and quantized predictions to determine weight sensitivity and maintains outlier features at higher precision levels to mitigate loss. GPTQ (Frantar et al., 2023) applies layer-wise quantization updating weights with inverse Hessian information. AWQ (Lin et al., 2024) stores the most impactful weights at high precision and determines scaling with per-channel methods. SqueezeLLM (Kim et al., 2024) uses k-means clustering for quantized weight values and stores sensitive weights sparsely.

## B    DETAILS OF UNLEARNING METHODS AND REGULARIZERS

We evaluate six efficient unlearning methods belonging to two distinct families of unlearning algorithms. We will first introduce these two families, which form the basis for the methods assessed. We will then discuss two regularizers addressing the lack of explicit design for utility preservation in these unlearning algorithms.

### B.1 Two Unlearning Families

**Gradient Ascent (GA)** minimizes the likelihood of correct predictions on $\mathcal{D}_{\text{forget}}$ by performing gradient ascent on the cross-entropy loss (Jang et al., 2023; Ilharco et al.; Yao et al., 2023). This method simply reverses the original training objective of minimizing the negative log-likelihood of the token sequences:

$$\min_\theta \mathcal{L}_{\text{GA}}(\theta) = \mathbb{E}_{x \sim \mathcal{D}_{\text{forget}}} \left[ \log(f_\theta(x_t | x_{<t})) \right] \tag{7}$$

where $f_\theta$ refers to the model parameterized by $\theta$ for unlearning, $x_{<t}$ represents the token sequence $x = (x_1, \ldots, x_{t-1})$, and $f_\theta(x_t | x_{<t})$ is the conditional probability that the LLM $f_\theta$ predicts $x_t$ given the preceding tokens $x_{<t}$.

**Negative Preference Optimization (NPO)** (Zhang et al., 2024) views the forget set as negative preference data and adapts the offline DPO objective (Rafailov et al., 2024) to fine-tune the model. This tuning ensures the model assigns a low likelihood to the forget set while remaining close to the original model $f_{\text{target}}$. The loss function for NPO is defined as:

$$\min_\theta \mathcal{L}_{\text{NPO}}(\theta) = -\frac{2}{\beta} \mathbb{E}_{x \sim \mathcal{D}_{\text{forget}}} \left[ \log \left( \sigma \left( -\beta \log \frac{f_\theta(x)}{f_{\text{target}}(x)} \right) \right) \right], \tag{8}$$

where $\sigma$ is the sigmoid function, and $\beta$ is a hyperparameter controlling the divergence of $f_\theta$ from $f_{\text{target}}$. We set $\beta = 0.1$ following the protocols in (Rafailov et al., 2024; Zhang et al., 2024).

### B.2 Utility Preservation through Regularization

GA and NPO are not explicitly designed for utility preservation. Hence, we explore regularization strategies to enhance performance on the retain set and ensure proximity to the target model during the unlearning process. These strategies include Gradient Descent on the Retain Set (GDR) and KL Divergence Minimization on the Retain Set (KLR):

**Gradient Descent on the Retain Set (GDR)** (Liu et al., 2022; Maini et al., 2024; Zhang et al., 2024) integrates a standard gradient descent objective on the cross-entropy of the retain set $\mathcal{D}_{\text{retain}}$. This approach is aimed at directly training the model to maintain its performance on $\mathcal{D}_{\text{retain}}$, aligning the unlearning objective with performance retention:

$$\min_\theta \mathcal{L}_{\text{GDR}} = \mathcal{L}_{\text{unlearn}} - \mathbb{E}_{x \sim \mathcal{D}_{\text{retain}}} \left[ \log(f_\theta(x_t | x_{<t})) \right] \tag{9}$$

where $\mathcal{L}_{\text{unlearn}}$ is a selected unlearning family.

**KL Divergence Minimization on the Retain Set (KLR)** (Maini et al., 2024; Zhang et al., 2024) aims to minimize the Kullback-Leibler (KL) divergence between the predictions on $x \in \mathcal{D}_{\text{retain}}$ of the unlearned model's probability distribution $p_{f_{\text{unlearn}}}(\cdot | x)$ over the vocabulary and the original model's probability distribution $p_{f_{\text{target}}}(\cdot | x)$ while maintaining the conventional unlearning loss on $\mathcal{D}_{\text{forget}}$. The formal objective can be written as:

$$\min_\theta \mathcal{L}_{\text{KL}} = \mathcal{L}_{\text{unlearn}} + \mathbb{E}_{x \in \mathcal{D}_{\text{retain}}} \text{KL}(p_{f_{\text{unlearn}}}(\cdot | x), p_{f_{\text{target}}}(\cdot | x)) \tag{10}$$

We integrate the GA and NPO methods with two regularizers, creating six unlearning methods in total: `GA`, `GA_GDR`, `GA_KLR`, `NPO`, `NPO_GDR`, and `NPO_KLR`.

## C Details of Evaluation Benchmark and Metrics

### C.1 NEWS and BOOKs Datasets

MUSE (Shi et al., 2024b) is a benchmark specifically developed for assessing LLM unlearning. It consists of two distinct datasets, **NEWS** and **BOOKS**, which focus on different types of textual data, i.e., news articles and books.

- **NEWS** (Li et al., 2023b) features a collection of BBC news articles from post-August 2023. These articles are systematically categorized into separate forget, retain, and holdout sets.

- **BOOKS** (Eldan & Russinovich, 2023) includes the entire Harry Potter series. The forget set comprises the original novels, whereas the retain set includes related materials from the Harry Potter FanWiki [1], ensuring retention of domain-specific knowledge post-unlearning.

## C.2 FOUR METRICS

Upon removing a forget set from a model, data owners expect the unlearned model to have (1) no Verbatim Memorization, (2) no Knowledge Memorization, and (3) no Privacy Leakage. Additionally, model deployers should note that removing specific data points can degrade model performance unpredictably, emphasizing the need for (4) utility preservation in the retain set. As such, following (Shi et al., 2024b), we consider four key metrics:

**Metric 1. VerMem: No Verbatim Memorization** (Jang et al., 2023; Eldan & Russinovich, 2023; Maini et al., 2024; Shi et al., 2024b). When a model no longer retains a sample or specific piece of information, it should not reproduce its contents exactly. We evaluate this type of verbatim memorization, known as VerbMem, by providing the model with the initial $l$ tokens of a sequence $x_{[1:l]}$ from $\mathcal{D}_{\text{forget}}$. We then compare the model's continuation output $f$ to the actual subsequent tokens $x_{[l+1:]}$ in $\mathcal{D}_{\text{forget}}$. This comparison uses the ROUGE-L F1 score (Lin, 2004) to quantify the degree of memorization.

$$\text{VerbMem}(f, \mathcal{D}) := \frac{1}{|\mathcal{D}_{\text{forget}}|} \sum_{x \in \mathcal{D}_{\text{forget}}} \text{ROUGE}(f(x_{[1:l]}), x_{[l+1:]}). \tag{11}$$

where a lower VerbMem value corresponds to better unlearning of verbatim memorization.

**Metric 2. KnowMem on $\mathcal{D}_{\text{forget}}$: No Knowledge Memorization** (Maini et al., 2024; Shi et al., 2024b). When a model has effectively unlearned a record or specific information, it should be unable to answer related questions. To evaluate the model $f$'s memorization of unlearned knowledge, we utilize the forget set $\mathcal{D}_{\text{forget}}$ formatted as question-answer pairs, following (Shi et al., 2024b). We first divide the original text into excerpts and use GPT-4 (Achiam et al., 2023) to create a question-answer pair $(q, a) \in GenQA(\mathcal{D}_{\text{forget}})$ for each excerpt. Next, we collect the model's responses to these questions, which are represented by $f(q)$. The average ROUGE score across all pairs in $\mathcal{D}_{\text{forget}}$ is computed to derive the knowledge memorization score:

$$\text{KnowMem}(f, \mathcal{D}_{\text{forget}}) := \frac{1}{|\text{GenQA}(\mathcal{D}_{\text{forget}})|} \sum_{(q,a) \in \text{GenQA}(\mathcal{D}_{\text{forget}})} \text{ROUGE}(f(q), a). \tag{12}$$

A lower KnowMem score signifies more successful unlearning and less residual knowledge memorization.

**Metric 3. PrivLeak: No Privacy Leakage** (Thudi et al., 2022; Maini et al., 2024; Shi et al., 2024b). The model that has unlearned information should not reveal whether $\mathcal{D}_{\text{forget}}$ is part of $\mathcal{D}_{\text{train}}$. Membership inference attacks (MIA) leverage statistical differences, such as next-token loss in LLMs to detect if an example is in the training set; a low loss suggests usage in training. Unlearning usually increases the loss of an example. Nevertheless, unlearning might not prevent privacy leaks if (1) the increase in loss is too small (under-unlearning) or (2) the loss is excessively high (over-unlearning). We use the Min-K% Prob method by (Shi et al., 2024a), a sophisticated MIA technique for LMs based on loss, and calculate the standard AUC-ROC score to distinguish members of $\mathcal{D}_{\text{forget}}$ from non-members in $\mathcal{D}_{\text{holdout}}$. By comparing the AUC score with that of the retrained model, privacy leakage is defined as:

$$\text{PrivLeak} := \frac{\text{AUC}(f_{\text{unlearn}}, \mathcal{D}_{\text{forget}}, \mathcal{D}_{\text{holdout}}) - \text{AUC}(f_{\text{retrain}}, \mathcal{D}_{\text{forget}}, \mathcal{D}_{\text{holdout}})}{\text{AUC}(f_{\text{retrain}}, \mathcal{D}_{\text{forget}}, \mathcal{D}_{\text{holdout}})}, \tag{13}$$

where an ideal unlearning algorithm produces a PrivLeak close to zero, indicating no privacy risk. Significant positive or negative PrivLeak values suggest over or under-unlearning. Generally, the AUC($f_{\text{retrain}}, \mathcal{D}_{\text{forget}}, \mathcal{D}_{\text{holdout}}$) value is approximately 0.5. However, intrinsic distribution shifts between $\mathcal{D}_{\text{forget}}$ and $\mathcal{D}_{\text{holdout}}$ can occasionally skew this away from 0.5.

**Metric 4. KnowMem on $\mathcal{D}_{\text{retain}}$: Utility Preservation** (Maini et al., 2024; Shi et al., 2024b). Since training models can be costly, an unlearning algorithm must maintain the model's performance on

---

[1] harrypotter.fandom.com/wiki

the retain set $\mathcal{D}_{\text{retain}}$. We measure the performance of the unlearned model on the retain set using the knowledge memorization metric $\text{KnowMem}(f, \mathcal{D}_{\text{retain}})$ in Eq. 12.

### C.3 DETAILS OF RETRAINED MODEL AND TARGET MODELS

Following the experimental setup in MUSE benchmark (Shi et al., 2024b), we use their open-sourced models. MUSE start with a general pretrained base model $f_0$, and finetune two models: $f_{\text{target}}$ on $\mathcal{D}_{\text{forget}} \cup \mathcal{D}_{\text{retain}}$, and $f_{\text{retrain}}$ on $\mathcal{D}_{\text{retain}}$ only. MUSE ensure that $f_0$ has no access to $\mathcal{D}_{\text{forget}}$ and $\mathcal{D}_{\text{retain}}$. Therefore, for NEWS, MUSE use $f_0 = \text{LLaMA-2 7B}$ (Touvron et al., 2023), which was released *before* the BBC news articles they use to construct the benchmark; and for BOOKS, MUSE use $f_0 = \text{ICLM-7B}$ (Shi et al., 2023), which does *not* contain the Harry Potter books in its pretraining data. Both models are finetuned for 5 epochs with a constant learning rate of $1e^{-5}$.

## D IMPLEMENTATION DETAILS

### D.1 IMPLEMENTATION DETAILS OF TABLE 1

Following the experimental setup in (Shi et al., 2024b), we implement six unlearning methods: GA, GA_GDR, GA_KLR, NPO, NPO_GDR, and NPO_KLR, using the AdamW optimizer (Loshchilov et al., 2017) with a fixed learning rate of $1e^{-5}$. We conduct the experiments over 10 and 5 epochs for the NEWS and BOOKS datasets, respectively. A grid search across $\{2, 5, 10, 100, 300\}$ determines the optimal weight $\alpha$ for the utility constraint on the retain dataset to balance unlearning performance with model utility. Table 4 shows the weight for regularization on the retain dataset for each method.

Table 4: Optimal regularization weights for each unlearning method.

| Unlearning Method | NEWS | BOOKS |
|---|---|---|
| GA_GDR | 1 | 100 |
| GA_KLR | 1 | 2 |
| NPO_GDR | 1 | 300 |
| NPO_KLR | 1 | 2 |

### D.2 IMPLEMENTATION DETAILS OF TABLE 3

The implementation of the original methods follows Appendix D.1. The detailed hyperparameter selection for the unlearning methods incorporating SURE is presented in Table 5.

Table 5: Optimal hyperparameters for each unlearning method.

| Unlearning Method | lr | $\alpha$ | $\gamma$ |
|---|---|---|---|
| GA_GDR + SURE | 1e-4 | 400 | Percentile($s$, 99) |
| GA_KLR + SURE | 1e-4 | 20 | Percentile($s$, 90) |
| NPO_GDR + SURE | 1e-4 | 300 | Percentile($s$, 99) |
| NPO_KLR + SURE | 1e-4 | 20 | Percentile($s$, 90) |

### D.3 IMPLEMENTATION DETAILS OF TABLE 8

The implementation of the original methods follows Appendix D.1. The detailed hyperparameter selection for the unlearning methods incorporating SURE is presented in Table 6.

## E ADDITIONAL EXPERIMENTAL ANALYSIS

In Table 1, we report on experiments involving the original model target $f_{\text{target}}$, the retrained model $f_{\text{retrain}}$, various unlearning methods, and their subsequent quantization at 8-bit and 4-bit precision using round-to-nearest (RTN). We compare these quantized models' unlearning performance to that of full-precision models. We exclude 2-bit precision models from testing due to their significant

Table 6: Optimal hyperparameters for each unlearning method.

| Unlearning Method | lr | $\alpha$ | $\gamma$ |
|---|---|---|---|
| GA_GDR + SURE | 1e-4 | 50 | Percentile($s$, 95) |
| GA_KLR + SURE | 1e-4 | 10 | Percentile($s$, 90) |
| NPO_GDR + SURE | 1e-4 | 50 | Percentile($s$, 95) |
| NPO_KLR + SURE | 1e-4 | 10 | Percentile($s$, 90) |

performance gap relative to full-precision models, which could distort interpretations of unlearning performance (Zhu et al., 2023). We observe the following: (1) Comparing $f_{\text{forget}}$ and $f_{\text{retrain}}$, the retrained model retains some knowledge of the forget set; it does not completely forget everything. (2) In the results for enhancements by GDR and KLR on metric M3, they represent two extremes. GDR explicitly performs gradient descent, resulting in lower losses and extremely positive results in privacy leaks. (3) Comparing GA and NPO with $f_{\text{target}}$ and $f_{\text{retrain}}$, both generally fail to achieve true forgetting due to poor performance on metrics M3 and M4. (4) Compared to GA and NPO, GA+X and NPO+X show worse performance on the privacy leakage metric M3, even though they perform well on the utility metric M4, suggesting that regularization helps preserve utility but not privacy. (5) Comparing $f_{\text{target}}$ with its quantized versions, there is a slight performance drop at 8-bit and a substantial drop at 4-bit, indicating that 4-bit quantization has a greater impact than 8-bit. (6) There is some performance loss in the 4-bit quantized version of $f_{\text{retrain}}$, but it is not pronounced. (7) Comparing all unlearned models with their 8-bit and 4-bit quantized versions, the 8-bit versions generally maintain performance comparable to the original across metrics M1, M2, M3, and M4. However, the 4-bit versions perform poorly; for example, in GA_KLR, M1 deteriorates from 14.1 to 33.8 and M2 from 27.1 to 50.9. Conversely, performance on M4 improves because the failure to unlearn is effectively closer to $f_{\text{target}}$, paradoxically indicating poorer unlearning.

# F  ADDITIONAL EVIDENCE OF THE FAILURE OF UNLEARNING VIA QUANTIZATION

In this section, we present empirical results on another LLM unlearning benchmark, RWKU (Jin et al., 2024), to demonstrate the generality of the issue of knowledge recovery through quantization. The data source for this benchmark consists of a list of famous individuals scraped from The Most Famous All-Time People Rank. These entities are linked to Wikipedia, and their popularity is measured using Wikipedia page views. The top 200 most popular entities, based on page view rankings, are selected as unlearning targets.

Specifically, we follow the experimental setup described in Table 1 of RWKU and report results on forgetting knowledge from the forget set. We evaluate four unlearning methods: **GA**, **NPO**, and **RT**. RT involves having the model generate questions related to the unlearning targets and then replacing its responses with "I do not know the answer." We use this refusal data to fine-tune the model so that it learns to reject questions related to the target.

We adopt three metrics:—**FB**, **QA**, and **AA** to measure the knowledge memorization of the unlearned model. Specifically, FB refers to fill-in-the-blank probes to examine the memory of the original training data related to the unlearning targets. QA assesses the ability of the unlearned model to utilize knowledge in practical applications through question-answer probes. Finally, AA involves more rigorous adversarial attack probes to evaluate unlearning efficacy. For all three metrics, lower values indicate better forgetting performance. We report the results in Table 7. From the table, we observe that for all unlearned models, the forgetting performance generally worsens after quantization, except for the unlearning method RT. In this case, the quantized model shows better forgetting performance in the FB metric. However, this improvement is due to the fact that the corresponding full-precision unlearned model retains a similar level of knowledge memorization as the target model. The improvement in forgetting performance is actually caused by a drop in model utility as a result of quantization. Overall, the results further reinforce the generality of the issue identified in this paper.

Table 7: Comparison of unlearning performance between full-precision and quantized models on RWKU benchmark.

| Method | Forget Set ↓ | | |
|---|---|---|---|
| | FB | QA | AA |
| Target $f_{target}$ | 85.9 | 76.4 | 77.7 |
| GA Full | 46.1 | 27.6 | 51.2 |
| GA Full + `Quan.(4 bit)` | 64.6 | 49.3 | 68.6 |
| NPO Full | 46.2 | 36.1 | 31.8 |
| NPO Full + `Quan.(4 bit)` | 48.9 | 42.7 | 44.0 |
| RT Full | 76.6 | 25.7 | 34.2 |
| RT Full + `Quan.(4 bit)` | 66.6 | 47.3 | 66.9 |

## G  EXPERIMENT RESULTS OF SURE ON NEWS DATASET

In this section, we report the results of SURE on the NEWS dataset; the experimental settings are outlined in Sec 6.2. The results are shown in Table 8. From the table, we observe results similar to those from experiments on the BOOKS dataset. Specifically: (i) for full-precision models, incorporating SURE into various unlearning approaches generally achieves similar forgetting performance and utility as the original methods; and (ii) for quantized models, SURE significantly improves forgetting performance compared to methods without it.

Table 8: Results of SURE on NEWS dataset.

| Method | Forget | | | Utility ↑ | | | | |
|---|---|---|---|---|---|---|---|---|
| | M1 ↓ | M2 ↓ | M3 → 0 | M4 | Gen | Tru | Fac | Flu |
| Target $f_{target}$ | 58.4 | 63.9 | -99.8 | 55.2 | 41.5 | 39.0 | 12.6 | 617.2 |
| GA_GDR | 0.0 | 28.9 | 87.1 | 34.2 | 38.0 | 36.5 | 11.2 | 562.0 |
| GA_GDR + SURE | 23.5 | 38.5 | -96.3 | 28.4 | 35.7 | 33.8 | 11.5 | 643.2 |
| GA_GDR + SURE + `Quan.` | 21.2 | 34.6 | -96.4 | 32.0 | 34.5 | 32.3 | 10.7 | 660.8 |
| GA_KLR | 14.1 | 27.1 | -91.6 | 23.1 | 33.3 | 40.3 | 12.1 | 560.6 |
| GA_KLR + SURE | 19.6 | 32.3 | -97.2 | 36.5 | 33.9 | 34.6 | 15.1 | 445.6 |
| GA_KLR + SURE + `Quan.` | 19.3 | 34.6 | -97.2 | 32.5 | 33.9 | 36.4 | 13.5 | 557.1 |
| NPO_GDR | 0.3 | 46.1 | 107.2 | 38.6 | 44.4 | 39.5 | 11.3 | 661.6 |
| NPO_GDR + SURE | 23.4 | 38.5 | -99.5 | 33.7 | 35.1 | 35.7 | 10.5 | 667.8 |
| NPO_GDR + SURE + `Quan.` | 21.1 | 35.9 | -99.6 | 32.0 | 34.5 | 37.4 | 10.0 | 669.0 |
| NPO_KLR | 16.6 | 36.6 | -94.0 | 33.3 | 34.5 | 41.6 | 11.7 | 539.8 |
| NPO_KLR + SURE | 19.4 | 31.5 | -98.8 | 35.9 | 38.6 | 35.3 | 9.9 | 458.1 |
| NPO_KLR + SURE + `Quan.` | 19.1 | 29.3 | -98.6 | 30.7 | 27.5 | 36.8 | 11.5 | 516.8 |

## H  HYPERPARAMETER ANALYSIS

In this section, we conduct hyperparameter analysis on the NEWS dataset. Since SURE introduces only one additional hyperparameter, $\gamma$, compared to the original unlearning methods, we focus solely on analyzing how $\gamma$ impacts unlearning performance. We report results for the full-precision model, as we find that all quantized versions of each method successfully prevent knowledge recovery through quantization. Therefore, we concentrate on the forgetting performance and model utility for the full-precision model. We follow the same experimental settings as in Appendix D.3. We choose the unlearning methods NPO_GDR and GA_KLR and set $\gamma = \text{Percentile}(s, x)$, where $x \in \{50, 90, 95, 99\}$. The results are presented in Table 9. From the table, we observe that increasing the value of $\gamma$ typically improves utility but degrades forgetting performance, with $\gamma = \text{Percentile}(s, 90)$ being a good threshold to achieve a trade-off.

## I  ABLATION STUDY

In this section, we present the results of the ablation study. Specifically, we aim to demonstrate that constructing a weight saliency map using the gradient of the loss $\mathcal{L}_{forget}$ with respect to the model weights on the forget dataset, and then updating only the salient weights, helps maintain model utility

Table 9: Hyperparameter analysis on NEWS dataset.

| Method | Forget | | | Utility ↑ | | | | |
|---|---|---|---|---|---|---|---|---|
| | M1 ↓ | M2 ↓ | M3 → 0 | M4 | Gen | Tru | Fac | Flu |
| Target $f_{target}$ | 58.4 | 63.9 | -99.8 | 55.2 | 41.5 | 39.0 | 12.6 | 617.2 |
| NPO_GDR + SURE ($\gamma$=Percent.($s$, 50)) | 24.5 | 39.4 | -99.7 | 32.5 | 33.7 | 34.5 | 8.4 | 644.7 |
| NPO_GDR + SURE ($\gamma$=Percent.($s$, 90)) | 23.9 | 44.7 | -99.7 | 38.4 | 36.8 | 35.9 | 9.0 | 658.8 |
| NPO_GDR + SURE ($\gamma$=Percent.($s$, 95)) | 23.4 | 38.5 | -99.5 | 33.7 | 35.1 | 35.7 | 10.5 | 667.8 |
| NPO_GDR + SURE ($\gamma$=Percent.($s$, 99)) | 23.4 | 40.9 | -99.7 | 39.8 | 38.6 | 38.3 | 9.5 | 672.6 |
| GA_KLR + SURE ($\gamma$=Percent.($s$, 50)) | 25.2 | 37.8 | -95.5 | 45.7 | 35.6 | 37.2 | 11.4 | 502.7 |
| GA_KLR + SURE ($\gamma$=Percent.($s$, 90)) | 25.2 | 37.8 | -95.5 | 45.7 | 35.7 | 37.2 | 11.5 | 524.8 |
| GA_KLR + SURE ($\gamma$=Percent.($s$, 95)) | 25.8 | 44.5 | -95.6 | 44.1 | 36.8 | 36.5 | 11.1 | 525.5 |
| GA_KLR + SURE ($\gamma$=Percent.($s$, 99)) | 24.8 | 46.3 | -95.5 | 44.7 | 37.6 | 39.3 | 13.3 | 530.1 |

and minimizes the potential bias caused by full fine-tuning with a large learning rate on the retain dataset. Therefore, we remove the saliency map construction module and refer to this version as SURE/S. We follow the experimental settings in Table 3 and conduct experiments on the BOOKS dataset. We set the learning rate as $1e^{-4}$. Hyperparameters are adjusted for each method to balance forgetting and model utility, ensuring a fair comparison. The results are shown in Table 10. From the table, we observe that SURE typically achieves a better balance between forgetting and utility, while SURE/S tends to forget more knowledge but performs worse in terms of model utility. This is because SURE/S fully fine-tunes the model with a large learning rate. The aggressive updates driven by the forgetting gradients can cause the model to over-adjust, leading to a decline in overall utility. Additionally, applying a large learning rate to the retain dataset can introduce a bias toward the retain data, potentially skewing the model's behavior and further degrading its performance on tasks outside the retain dataset.

Table 10: Ablation study on BOOKS dataset.

| Method | Forget | | | Utility ↑ | | | | |
|---|---|---|---|---|---|---|---|---|
| | M1 ↓ | M2 ↓ | M3 → 0 | M4 | Gen | Tru | Fac | Flu |
| Target $f_{target}$ | 99.8 | 59.4 | -57.5 | 66.9 | 28.7 | 33.6 | 9.1 | 573.3 |
| GA_GDR + SURE/S | 0.0 | 0.0 | -48.3 | 0.0 | 27.4 | 0.22 | 0.0 | 530.5 |
| GA_GDR + SURE | 0.0 | 0.3 | -6.4 | 49.3 | 29.2 | 0.2 | 0.0 | 544.9 |
| GA_KLR + SURE/S | 14.3 | 2.6 | -31.3 | 1.6 | 20.0 | 22.3 | 6.2 | 472.9 |
| GA_KLR + SURE | 16.6 | 25.3 | -57.9 | 46.5 | 22.8 | 28.6 | 9.7 | 546.7 |
| NPO_GDR + SURE/S | 0.0 | 10.9 | -51.4 | 54.2 | 22.2 | 27.3 | 3.0 | 414.2 |
| NPO_GDR + SURE | 0.0 | 31.2 | -48.2 | 46.1 | 25.2 | 39.5 | 7.3 | 505.9 |
| NPO_KLR + SURE/S | 3.6 | 0.0 | -31.4 | 0.0 | 21.0 | 23.2 | 0.0 | 368.2 |
| NPO_KLR + SURE | 17.6 | 37.8 | -58.0 | 49.4 | 23.4 | 30.2 | 7.4 | 588.8 |

## J EXPERIMENTS AND DISCUSSION ON OTHER UNLEARNING METHODS

Recent unlearning methods (Sheshadri et al., 2024; Li et al., 2024b) potentially encourage deeper forgetting in model representations. To validate the pervasiveness of the identified issue, we adopt the unlearning methods RMU (Li et al., 2024b) and LAT (Sheshadri et al., 2024) and conduct experiments on the BOOKS dataset. For LAT, we combine it with RMU to form the method RMU_LAT. We tune the learning rate using grid search in $[5e^{-6}, 1e^{-5}, 5e^{-5}]$ to balance unlearning performance and model utility. The results are shown in Table 11. It is evident that, for the unlearning metrics M1, M2, and M3, both methods show poor performance after quantization. Our results further highlight the challenge of balancing the prevention of knowledge recovery through quantization with maintaining model utility.

Task Vector is a representative non-finetuning-based unlearning method. However, as observed in (Dong et al., 2024; Barbulescu & Triantafillou, 2024), task vectors achieve poorer unlearning performance compared to GA, with the performance gap being significantly larger in the MUSE benchmark (Shi et al., 2024b). In contrast, GA, a finetuning-based unlearning method, demonstrates strong unlearning performance across all benchmarks. Thus, we acknowledge that non-finetuning-based unlearning techniques exist, which may not be vulnerable to quantization issues but generally exhibit lower unlearning performance overall.

Table 11: Unlearning performance on other unlearning methods

| Method | Forget | | | Utility ↑ | | | | |
|---|---|---|---|---|---|---|---|---|
| | M1 ↓ | M2 ↓ | M3 → 0 | M4 | Gen | Tru | Fac | Flu |
| Target | 99.8 | 59.4 | -57.5 | 66.9 | 28.7 | 33.6 | 9.1 | 573.3 |
| RMU | 14.2 | 0.0 | -28.3 | 0.0 | 29.8 | 36.5 | 0.0 | 69.3 |
| RMU + `Quan.` | 50.5 | 29.3 | -57.2 | 41.8 | 25.1 | 32.3 | 4.9 | 689.5 |
| RMU_LAT | 11.7 | 0.0 | -40.5 | 0.0 | 30.9 | 36.3 | 0.0 | 1.99 |
| RMU_LAT + `Quan.` | 43.0 | 27.4 | -57.6 | 37.9 | 25.7 | 36.9 | 4.8 | 685.9 |

## K  FURTHER INSIGHTS INTO THE CORE CONTRIBUTIONS

In this section, we provide further insights into the core contributions of our paper. The core contributions of our paper primarily lie in identifying a novel problem, providing a theoretical analysis, and making an initial effort to mitigate this issue while inspiring future research. Specifically,

- We identify a critical and interesting problem in LLM unlearning: the risk of knowledge recovery in unlearned models via quantization. We also conduct extensive experiments to verify the pervasiveness of this problem.

- We provide a theoretical analysis of the occurrence of the problem. Based on our theoretical analysis, we identify increasing weight changes as a promising approach that has been overlooked by existing unlearning methods. To empirically validate our analysis, we tested a method to enlarge the learning rate, and our results in Table 3 demonstrate that this approach shows promise in preventing knowledge recovery via quantization.

- However, as demonstrated in the experimental results in the ablation study (Appendix I), a large learning rate can lead to model performance degradation. Thus, we extend the concept of localization-informed unlearning methods to the domain of LLMs by calculating module-level gradients and constructing saliency maps. It is important to note that this is not the only approach to mitigating the negative impact of a large learning rate. Our aim here is to make an initial effort to empirically verify whether localization-informed unlearning could be a promising solution to help mitigate the effects of a large learning rate. Overall, our initial efforts validate the theoretical analysis and inspire future research aimed at increasing weight changes during unlearning to prevent knowledge recovery via quantization. We thank the reviewer for pointing this out and will include a discussion in a future version of our paper.

## L  THE CRITICAL NATURE AND REAL-WORLD SCENARIOS OF THE IDENTIFIED PROBLEM

Although the results in Table 1 show that the problem of knowledge recovery via quantization pervasively exists only in 4-bit quantization, 4-bit quantization is widely studied (Frantar et al., 2023; Lin et al., 2024), and there is even significant research focusing on lower-bit formats, such as 1-bit quantization (Xu et al., 2024b; Huang et al., 2024b). Quantization is a widely used technique for deploying LLMs in resource-constrained scenarios, and advanced quantization methods focusing on low-bit quantization are attracting increasing research interest. Our empirical results in Table 1 also demonstrate that the 4-bit model does not exhibit significant utility degradation compared to the full-precision version. Thus, we believe the identified issue of quantization leading to unlearning performance degradation will spark significant research interest within the community.

Many open-sourced LLMs, such as Llama (Touvron et al., 2023; Dubey et al., 2024), are available for users to download in full precision. Imagine a scenario where a data owner requests a company (model provider) to remove copyrighted data from their open-sourced model. The model provider would apply an unlearning algorithm to update their model. The data owner could then download the updated model and freely use various methods to test whether the model has truly unlearned the specified data. Meanwhile, quantization is a popular and widely used technique in the era of LLMs. Therefore, it is essential for the model provider to ensure robust unlearning to prevent knowledge

recovery through quantization. Failure to do so may expose the company to legal risks, as they could face charges from the data owner for non-compliance.

## M    COMPARISON WITH OTHER METHODS FOR EVALUATING ROBUST UNLEARNING IN LLMS

In this section, we prodive discussion on the difference between quantization and other methods in evaluating robust unleanring in LLMs:

- Our motivation is different from attacking the unlearned model and forcing it to recover the knowledge. Instead, we consider the problem from the perspective that, as quantization becomes increasingly popular in the era of LLMs, it is important to evaluate whether existing unlearning methods remain effective when the unlearned model is quantized.

- The performance of forcing an unlearned model to generate the supposed forgotten knowledge is poor (Lynch et al., 2024). Specifically, the unlearned model retains only 10% of the knowledge, and none of the attack methods (including Other Language, Jailbreaks, In-Context Extraction, Fine-Tuning, Downstream Task, Latent Knowledge, Prompting Baselines, Side Effects) achieve the goal of forcing the unlearned model to retain 20% of the knowledge. These results are much less effective compared to our empirical findings. Moreover, the attack methods mentioned in (Lynch et al., 2024), such as Jailbreak and In-context Learning, require significantly more effort to recover knowledge from an unlearned model compared to the simplicity and effectiveness of quantization.

## N    REPRODUCIBILITY

We provide experimental setup and implementation details in Appendix D. Our code is available at: https://github.com/zzwjames/FailureLLMUnlearning.

