# OpenReview forum: "Catastrophic Failure of LLM Unlearning via Quantization"
_ICLR.cc/2025/Conference — ICLR 2025 Poster_

### Official Review · Reviewer_Sbbj · 2024-10-19

**Soundness:** 2
**Presentation:** 3
**Contribution:** 2
**Rating:** 5
**Confidence:** 4

**Summary:**

This paper reveals the failure of LLM unlearning after model quantization. It presents a theoretical analysis of the reason why quantization weakens the unlearning effectiveness. A new unlearning framework with a larger learning rate and salient weight selection is proposed to alleviate the failure of LLM unlearning via quantization. Extensive experiments are conducted to consolidate the existence of claimed failure and the effectiveness of the proposed unlearning method.

**Strengths:**

- This paper reveals an interesting phenomenon -- unlearning failure brought by model quantization.
- The theoretical analysis of the phenomenon is persuasive.
- The paper is well-structured.

**Weaknesses:**

- I am not convinced that the "quantization attack" represents a practical scenario. Quantization is typically employed during training when computational resources are limited. However, companies do not need to quantize their models when deploying them in the cloud. Therefore, it is unlikely that an unlearned model would experience a "quantization attack" in real-world business applications.
- While claiming the work towards "catastrophic failure of LLM unlearning via quantization", some critical unlearning methods (e.g.,  task vector [1]) are ignored. The authors only investigate fine-tuning-based methods.
- The proposed method SURE is not impressive enough. It is a typical localization-informed unlearning method [2] and is widely adopted for achieving optimized unlearning-utility tradeoff [3,4]. Although the authors adopt this method to mitigate the loss of utility brought by a large learning rate, I am not convinced that it is an innovative contribution. Moreover, the equation 6 is highly similar as equation 4 in [3].
- This paper lacks some experiment details (e.g., backbone model).

[1] Ilharco G, Ribeiro M T, Wortsman M, et al. Editing models with task arithmetic[C]//The Eleventh International Conference on Learning Representations.

[2] Liu S, Yao Y, Jia J, et al. Rethinking machine unlearning for large language models[J]. arXiv preprint arXiv:2402.08787, 2024.

[3] Fan C, Liu J, Zhang Y, et al. SalUn: Empowering Machine Unlearning via Gradient-Based Weight Saliency in Both Image Classification and Generation[C]//International Conference on Learning Representations. 2024.

[4] Yu C, Jeoung S, Kasi A, et al. Unlearning bias in language models by partitioning gradients[C]//Findings of the Association for Computational Linguistics: ACL 2023. 2023: 6032-6048

**Questions:**

- I am wondering if enlarging the number of epochs achieves similar results as increasing the learning rate, especially for unbonded unlearning objective such as GA.
- What if the unlearning methods are directly applied to a quantized model? Will this operation help avoid the failure of unlearning?
- Why do you choose max|w|=200 as an example to explain how quantization affects unlearning?

---

> ### Author Response · Authors · 2024-11-20
>
> We thank the reviewer for acknowledging the novelty of the phenomenon we identified and the strength of our theoretical analysis. Below, we provide a point-by-point response to the reviewer's comments:
>
> **W1: practical scenario of quantization attack**
>
> Thank you for your insightful question. The reviewer's comments mentioned that "companies do not need to quantize their models when deploying them in the cloud." However, many open-sourced LLMs, such as Llama [1], are available for users to download in full precision. Imagine a scenario where a data owner requests a company (model provider) to remove copyrighted data from their open-sourced model. The model provider would apply an unlearning algorithm to update their model. The data owner could then download the updated model and freely use various methods to test whether the model has truly unlearned the specified data.
>
> Meanwhile, quantization is a popular and widely used technique in the era of LLMs. Therefore, it is essential for the model provider to ensure robust unlearning to prevent knowledge recovery through quantization. Failure to do so may expose the company to legal risks, as they could face charges from the data owner for non-compliance.
>
> We appreciate the reviewer for highlighting this point and will incorporate the discussion in a future revision of our paper.
>
> ---
>
>
> **W2: Missing unlearning method: task vector**
>
> We thank the reviewer for pointing this out. However, according to the results in MUSE [2], the unlearning method task vector achieves poor unlearning performance compared to the fine-tuning based methods in our paper. Specifically, on the NEWS dataset, the model still memorizes 97.9% of the knowledge after unlearning with the task vector, and on the BOOKS dataset, the model still memorizes 99.8% of the knowledge. Thus, we do not include the task vector, as our paper mainly focuses on methods that demonstrate strong unlearning performance in full-precision models.
>
>
> ---
>
>
> **W3: the proposed method not impressive**
>
> We acknowledge that localization-informed unlearning methods have been explored in several existing works, and we have appropriately cited the relevant studies in Section 6.1, where we introduced our approach.
> However, we would like to emphasize that the novelty and contribution of the proposed method lie in the deep insights that guided its design. The experimental results provide further insights for future research. The **core contributions of our paper primarily lie in identifying a novel problem, providing a theoretical analysis, and making an initial effort to mitigate this issue while inspiring future research**. Specifically:
>
> * Based on our theoretical analysis, **we identify increasing weight changes as a promising approach that has been overlooked by existing unlearning methods.**. To empirically validate our analysis, we tested a method to enlarge the learning rate, and our results in Table 3 demonstrate that this approach shows promise in preventing knowledge recovery via quantization.
> * However, as demonstrated in the experimental results in the ablation study (Appendix H), a large learning rate can lead to model performance degradation. Thus, we extend the concept of localization-informed unlearning methods to the domain of LLMs by calculating module-level gradients and constructing saliency maps. It is important to note that this is not the only approach to mitigating the negative impact of a large learning rate. **Our aim here is to make an initial effort to empirically verify whether localization-informed unlearning could be a promising solution to help mitigate the effects of a large learning rate**.
>
> Overall, our initial efforts validate the theoretical analysis and inspire future research aimed at increasing weight changes during unlearning to prevent knowledge recovery via quantization. We thank the reviewer for pointing this out and will include a discussion in a future version of our paper.
>
>
> ---
>
> **W4: more experimental details**
>
> Thanks for pointing this out. We have included additional experimental details in the Appendix C.3 of the updated version of our paper.
>
>
>
> [1] Llama 2: Open foundation and fine-tuned chat models. arXiv.

---

> ### Author Response · Authors · 2024-11-20
>
> **Q1: the effect of enlarging unlearning epochs**
>
> Thank you for your insightful question. Following your advice, we conducted experiments on the NEWS dataset with 1 epoch and 10 epochs, respectively. The results are shown in the table below. All models were quantized to 4-bit after unlearning. It is evident that increasing the unlearning epochs from 1 to 10 does not mitigate the issues we identified. This can be attributed to the fact that the model has already converged and successfully unlearned in full precision, and thus, it does not make significant updates after convergence.
>
> | Method    | Verm_f | Know_f | priv   | Know_r |
> |-----------|--------|--------|--------|--------|
> | GA (1 epoch)       | 0      | 0      | 29.9   | 0      |
> | GA (10 epoch)        | 0      | 0      | 24.5   | 0      |
> | GA_GDR (1 epoch)   | 33.5   | 55.5   | -99.6  | 48.1   |
> | GA_GDR (10 epoch)    | 25.0   | 50.1   | -99.1  | 47.7   |
> | GA_KLR (1 epoch)   | 35.6   | 52.3   | -99.8  | 46.5   |
> | GA_KLR (10 epoch)   | 33.8   | 50.9   | -99.8  | 45.8   |
> | NPO (1 epoch)       | 12.2   | 25.5   | -5.4   | 21.4   |
> | NPO (10 epoch)       | 16.2   | 25.4   | -71.6  | 27.9   |
> | NPO_GDR (1 epoch)   | 28.8   | 48.9   | -99.8  | 49.0   |
> | NPO_GDR (10 epoch)   | 33.2   | 51.4   | -99.8  | 48.2   |
> | NPO_KLR (1 epoch)   | 29.8   | 51.1   | -99.8  | 46.6   |
> | NPO_KLR (10 epoch)   | 34.1   | 53.7   | -99.8  | 48.8   |
>
>
>
> ---
>
> **Q2: unlearning methods directly applied to a quantized model**
>
> Thanks for your insightful question.
>
> * 4-bit quantization results in a model with parameters in an INT format. It is uncommon to fine-tune a quantized model since most optimizers in PyTorch (e.g., Adam, SGD) are designed to work with floating-point parameters.
> * Existing quantization methods are designed to minimize the quantization error caused by lower-bit parameters, for example, by adjusting the scale factor during the quantization process [3]. To the best of our knowledge, there is no existing work that focuses on fine-tuning a quantized model directly.
> * As mentioned earlier, we consider a scenario where a model provider releases their model online, and users are free to download it. In this case, when a data owner requests the model provider to remove the knowledge of their data, it would not be reasonable for the model provider to offer a fine-tuned quantized model to the data owner for testing.
>
>
> ---
>
> **Q3: why |w|=200**
>
> Thank you for your question. We acknowledge that a model weight value of 200 may be considered an outlier. In our paper, we use the example of |w|=200 to ensure that the quantization integer value is not too small, making the explanation clearer and more readable.
>
>
> [2] MUSE: Machine Unlearning Six-Way Evaluation for Language Models. arXiv.
>
> [3] AWQ: Activation-aware Weight Quantization for On-Device LLM Compression and Acceleration. MLSys 2024.

---

> ### Comment · Reviewer_Sbbj · 2024-11-24
> **Official Comment**
>
> I appreciate the authors for their reply. I still have some concerns about this paper.
>
> * Task vector is proven to be a strong baseline in the works about machine unlearning [1, 2]. Although it performs badly on the MUSE benchmark, it shows good effects on other datasets and should not be left out.
>
> * I am confused about the results of the additional experiments. Normally, we set a threshold to stop unlearning at some steps to preserve the general ability of the model. However, your results indicate that training the model for 10 epochs has a relatively little side-effect on the model ability which is counter-intuitive to me. I would suggest include learning curve in the paper to validate the claim "The model has already converged."
>
> Despite the concerns mentioned above, the authors address some of my concerns and I will increase my score a bit.
>
> [1] Dong Y R, Lin H, Belkin M, et al. Unmemorization in Large Language Models via Self-Distillation and Deliberate Imagination[J]. arXiv preprint arXiv:2402.10052, 2024.
>
> [2] Bărbulescu G O, Triantafillou P. To Each (Textual Sequence) Its Own: Improving Memorized-Data Unlearning in Large Language Models[C]//Forty-first International Conference on Machine Learning.

---

> > ### Author Response · Authors · 2024-11-25
> >
> > We sincerely thank the reviewer for the reply and for recognizing our efforts. Below, we provide a point-by-point response to the reviewer's remaining concerns:
> >
> > > Task vector is proven to be a strong baseline in other benchmarks
> >
> > We thank the reviewer for highlighting related work. As observed in [1][2], task vectors achieve poorer unlearning performance compared to GA, with the performance gap being significantly larger in the MUSE benchmark. In contrast, GA (finetuning-based unlearning method) demonstrates strong unlearning performance across all benchmarks. In the next version of our paper, we will include a discussion acknowledging that non-finetuning-based unlearning techniques exist, which may not be vulnerable to quantization issues but generally exhibit lower unlearning performance overall.
> >
> >
> >
> >
> > ---
> >
> >
> > > Further explanation of the results across different epochs.
> >
> > We agree with the reviewer that a higher number of epochs for unlearning would typically lead to a utility drop for the unlearned model. However, **our reported results are for the 4-bit quantized model**. During unlearning, when the full-precision model converges, it usually undergoes only small weight changes thereafter. In our experiments, the model converges after one epoch. Therefore, after 4-bit quantization, the performance remains similar for unlearning across different epochs.
> >
> > Thank you for your active discussion and constructive comments! We hope our responses address the reviewer’s remaining concerns, and we would greatly appreciate a raise to a positive rating.
> >
> >
> > ---
> >
> >
> > [1] Dong Y R, Lin H, Belkin M, et al. Unmemorization in Large Language Models via Self-Distillation and Deliberate Imagination[J]. arXiv preprint arXiv:2402.10052, 2024.
> >
> > [2] Bărbulescu G O, Triantafillou P. To Each (Textual Sequence) Its Own: Improving Memorized-Data Unlearning in Large Language Models[C]//Forty-first International Conference on Machine Learning.

---

> > ### Author Response · Authors · 2024-11-29
> >
> > We sincerely thank **Reviewer Sbbj** for the thoughtful review of our paper and for engaging in active discussions. Regarding the reviewer's last two concerns, we have added a discussion in Appendix I to address the concern regarding the task vector. Additionally, we offer the following response to the reviewer's final question on "whether larger training epochs lead to the same result as a large learning rate in achieving significant weight changes during unlearning."
> >
> > Firstly, **we would like to respectfully clarify that the question of whether larger training epochs lead to the same result as a large learning rate in achieving significant weight changes during unlearning does not impact the contributions of our paper**. In Section 6, to verify our theoretical analysis in Section 5—that larger weight changes can prevent knowledge recovery via quantization—we chose a large learning rate for its intuitive and straightforward nature. We do not claim that a large learning rate is the only option for achieving significant weight changes. Our primary aim is to validate our theoretical analysis and provide an initial empirical demonstration that a larger learning rate can prevent knowledge recovery through quantization, potentially inspiring future research.
> >
> > Given the reviewer's interest in this question, **we made significant efforts to provide a detailed and comprehensive experimental comparison in our initial rebuttal**, empirically demonstrating that larger epochs may fail to achieve the same effect as a larger learning rate. We also **provided detailed explanations to support this conclusion**. Due to time constraints, we were unable to include the training loss curve in the revised paper.
> >
> > Returning to the question, our goal is to ensure that the weight changes exceed a threshold so that the unlearning results can be effectively projected onto the 4-bit quantized model. Thus, a larger learning rate is the most direct and intuitive method to achieve significant weight changes.
> >
> > In the initial review, the reviewer raised the idea of "larger epochs for unbounded objectives, e.g., GA." However, effective unlearning requires the model to forget specific knowledge while preserving its utility. Consequently, we do not consider GA without any constraints, as both larger learning rates and more epochs can lead to a drastic drop in model utility.
> >
> > **The reasons larger epochs may fail to produce significant weight changes are as follows:**
> >
> > * Nature of the training process: During training, the optimizer navigates the loss landscape to minimize the loss function. This landscape is often complex, with numerous local minima and flat regions (plateaus). When the optimizer reaches a local minimum, without an appropriate learning rate, the gradients become very small, resulting in small parameter updates.
> >
> > * Local minima near the original model: The effect of additional epochs heavily depends on the weight ratio between the utility constraint and the forgetting loss. For successful unlearning, this weight ratio must be set such that the model can unlearn the knowledge with minimal weight changes to maintain utility. Consequently, during effective unlearning, the model tends to converge to local minima near the original model parameters.
> >
> > We will include a discussion on the use of larger epochs in the final version of the paper.
> >
> > With the discussion deadline approaching, we would greatly appreciate it if the reviewer could kindly review our responses at the earliest convenience. We sincerely hope that our rebuttal has fully addressed the reviewer’s concerns and would be truly grateful if the reviewer might consider raising the evaluation to a positive rating.

---

> ### Comment · Reviewer_Sbbj · 2024-12-01
> **Official Comment**
>
> I would like to thank the authors for the explanation.
> First, I would like to highlight that the task vector demonstrates performance comparable to NPO, while GA fails entirely to preserve model utility, as shown in [1]. This indicates that the task vector is a more advanced unlearning method than GA, as it achieves a better balance between unlearning and model utility. Therefore, if GA is used as a baseline, it would be natural to also compare it with the task vector.
>
> Second, the reason why I ask the authors to provide the training curve during unlearning is that the large number of epochs is expected to destroy the model, making the model generate nonsensical or repetitive outputs. The additional results from the authors suggest that the model still keeps its utility after training 10 epochs (which is a large number of epochs for unlearning I believe), which is counter-intuitive to me. I understand the result is about the unlearned quantified model. However, it is hard to be convinced that quantization would recover a collapsed model.
>
> Based on the reasons above, I will maintain my score.
>
> [1] Dong Y R, Lin H, Belkin M, et al. Unmemorization in Large Language Models via Self-Distillation and Deliberate Imagination[J]. arXiv preprint arXiv:2402.10052, 2024.

---

> ### Author Response · Authors · 2024-12-02
>
> We sincerely thank the reviewer for active discussion and valuable feedback.
>
> > task vector
>
> We appreciate the reviewer for pointing out the task vector and non-finetuning-based methods.
> * In the MUSE benchmark, the task vector demonstrated extremely poor performance in unlearning, which is why we did not include it in our tests.
> * Although GA itself can lead to a drastic drop in performance, when combined with utility constraints, it typically can achieve better results in balancing unlearning performance and model utility [1]. And the main claim of our paper is that quantization can result in a drop in unlearning performance for methods that incorporate utility constraints. Furthermore, existing SOTA unlearning methods primarily focus on finetuning-based approaches, which remain more widely studied compared to non-finetuning-based methods [2].
> * According to the reviewer's suggestion, we will add a discussion on non-finetuning-based methods and further clarify that our primary focus is on finetuning-based methods in the final version.
>
> > experiments on large epochs
>
> **The core reason the model does not collapse is that we have a utility constraint during unlearning.**
>
> We respectfully agree with the reviewer that larger epochs tend to lead utility drop. However, during unlearning, training epochs is not the only factor that influences the balance between unlearning performance and model utility. **Learning rate and utility constraints also play a critical role**. As described in Appendix D, in our experiments, **we used a small learning rate with a peak value of 1e-5 and maintained utility constraints to ensure the model retains utility during unlearning**. (We omit the discussion on GA and NPO here, as their utility is inherently poor.)
>
> To fully address your concern, we choose unleanring method GA_KLR and **present the utility performance of the full-precision unlearned model across different epochs in the following table.**
>
> | Epoch    | M4     | Gen   | Tru   | Fac   | Flu   |
> |----------|--------|-------|-------|-------|-------|
> | Epoch 1  | 55.4   | 26.9  | 35.7  | 4.5   | 659.8 |
> | Epoch 2  | 54.1   | 30.4  | 35.6  | 4.5   | 682.8 |
> | Epoch 3  | 53.6   | 26.3  | 35.5  | 3.8   | 672.2 |
> | Epoch 4  | 53.9   | 30.4  | 35.6  | 4.1   | 672.5 |
> | Epoch 5  | 54.9   | 24.6  | 35.8  | 4.2   | 670.7 |
> | Epoch 6  | 54.09  | 26.3  | 35.8  | 3.7   | 673.5 |
> | Epoch 7  | 55.8   | 27.3  | 3.6   | 3.5   | 685.5 |
> | Epoch 8  | 54.4   | 22.8  | 36.3  | 3.5   | 678.5 |
> | Epoch 9  | 53.1   | 27.5  | 35.9  | 3.2   | 687.2 |
> | Epoch 10 | 54.5   | 24.6  | 36.2  | 3.1   | 691.0 |
>
>
> From the table, we observe that the model utility remains stable across five utility metrics. Combined with the unlearning performance presented in the initial rebuttal, **the results indicate that we have properly configured the learning rate and the weight for the utility constraint, ensuring that the model can effectively unlearn the knowledge while maintaining utility even with large training epochs, thereby addressing the reviewer's concerns**. We have made our code available and warmly invite the reviewer to replicate our results. We sincerely appreciate the reviewer's thoughtful feedback and recognize there may have been a misunderstanding regarding our focus. **The balance between unlearning performance and model utility is inherently adjustable by modifying training epochs, learning rate, and the weights for the utility constraint.** We believe this does not diminish the contributions of our paper, as the learning rate and the utility constraint weights can be carefully adjusted to achieve the desired results if the reviewer anticipates that a larger training epoch may lead to greater weight changes or model collapse.
>
> We sincerely thank the reviewer for dedicating time to review our paper and for engaging in thoughtful discussions. We truly hope the above response helps to address the reviewer's concerns.
>
> [1] MUSE: Machine Unlearning Six-Way Evaluation for Language Models. arXiv Jul 2024.
>
> [2] Rethinking Machine Unlearning for Large Language Models. arXiv Feb 2024.

---

### Official Review · Reviewer_KTWe · 2024-11-03

**Soundness:** 3
**Presentation:** 1
**Contribution:** 3
**Rating:** 6
**Confidence:** 3

**Summary:**

This paper highlights a significant vulnerability in current LLM unlearning methods, showing that quantization can often restore unlearned knowledge. This issue arises because unlearning only results in minor weight changes. The authors investigate this phenomenon using various quantization methods and precision levels and then propose a saliency-based module selection approach during unlearning training as a potential solution. The experiment result show that this is effective.

**Strengths:**

* The paper reveals a common issue in existing LLM unlearning methods, providing valuable insights for the field.
* Comprehensive experiments clearly demonstrate the problem, and the proposed solution shows effectiveness in mitigating it

**Weaknesses:**

* In Section 5, the authors attribute the failure of existing unlearning methods to the minimal difference between the weights of the unlearned LLM and the original LLM. However, this claim lacks direct experimental evidence. A simple cosine similarity or L2 norm comparison between the two sets of parameters would help validate this explanation.
* The paper’s presentation can be largely improved.
	* Currently, I think the writing is too wordy; for example, the phrase "we will explain the cause of these observations in Section 5" is repeated multiple times (lines 298, 313), which interrupts the flow. I think moving Section 5 earlier or reducing the lengthy experimental setup in Section 4 could improve readability.
	* Replacing M1/2/3/4 with the actual metric names instead of M1/2/3/4 in Tables 1-4 would help readers better understand the table.

* Minor:
  * table 4 seems to exceed the page width
  * It would be better if the names of different quantization methods could be denoted using different font families, such as textsc/texttt. This can better discriminate between other texts and help reader understand.

**Questions:**

* Could you clarify the process for selecting the saliency module? Line 431 references the weights of attention heads or feedforward layers, but it's not clear the actual modules considered in training.
* During SURE training, which modules are actually updated? Is there a possibility that most modules remain unchanged? Is it possible that the modules updated at the early stage of training and later stage are different? This could provide deeper insights into how knowledge is stored within an LLM.

---

> ### Author Response · Authors · 2024-11-21
>
> We thank the reviewer for acknowledging the importance and novelty of the problem identified, as well as the comprehensive experiments presented in our paper. Below, we provide a point-by-point response to the reviewer's comments:
>
>
> ---
>
>
> **W1: empirical results for Section 5**
>
> Thank you for your valuable suggestion. Following your advice, we conducted experiments on the BOOKS dataset to (1) calculate the cosine similarity between $f_{target}$ and unlearned models using different techniques, and (2) calculate the percentage of weights with different values between $f_{target}$ and the unlearned models. All models are 4-bit quantized. The results are shown in the table below. It is evident that all unlearned models show small differences compared to the target model without unlearning.
>
> | Model    | Cosine | Discrepancy |
> |----------|--------|-------------|
> | ga_klr   | 0.99  | 0.67%       |
> | ga_gdr   | 0.99  | 0.60%       |
> | npo_klr  | 0.99  | 0.67%       |
> | npo_gdr  | 0.99  | 0.64%       |
>
>
> ---
>
>
> **W2 & W3: minors points on writing**
>
> We thank the reviewer for reading our paper in detail and for pointing out these minor issues. We will address them in the future version of our paper.
>
>
>
> ---
>
>
> **Q1: the process of selecting the saliency module**
>
> During the unlearning process, we update the self-attention module, feedforward network, and layer normalization layers. Specifically, for each module, we sum the gradient magnitudes of the forgetting loss within the module and then set a threshold to select the modules with large gradient magnitudes for updating.
>
>
> ---
>
>
> **Q2:**
> > During SURE training, which modules are actually updated?
>
> We propose updating the self-attention module, feedforward network, and layer normalization layers.
>
> > Is there a possibility that most modules remain unchanged?
>
> In our paper, we manually set a threshold to selectively update modules that exhibit large gradients on the forgetting loss. Typically, we set a threshold that updates only a small part of the model. As a result, most modules remain unchanged.
>
> > Is it possible that the modules updated at the early stage of training and later stage are different?
>
> According to our empirical observations, the modules updated during the unlearning process typically lie in the later layers of the LLM, which is consistent with the findings in [1].
>
>
> ---
>
>
> [1] LLMs Know More Than They Show: On the Intrinsic Representation of LLM Hallucinations. arXiv Oct 2024.

---

> > ### Comment · Reviewer_KTWe · 2024-11-21
> >
> > Thanks for the response. I would maintain my score.

---

> > > ### Author Response · Authors · 2024-11-24
> > >
> > > We thank the reviewer for valuable comments and support for our paper. We are happy to address any further questions.
> > >
> > > Best regards,
> > >
> > > The Authors

---

### Official Review · Reviewer_ZUW5 · 2024-11-04

**Soundness:** 3
**Presentation:** 2
**Contribution:** 2
**Rating:** 6
**Confidence:** 4

**Summary:**

The paper explores the effect of weight quantization on models tuned via machine unlearning towards intentionally forgetting and hiding knowledge on copyrighted or private data. Interestingly, results reveal that 4-bit quantization of model weights can restore "forgotten" information from models unlearned using methods such as gradient ascent (GA) or negative preference optimization (NPO). The authors intuitively explain this is due to weight changes from unlearning being too small, and propose SURE, a novel weight saliency-based unlearning technique that allows use of large learning rates while minimizing potential biases towards small retain sets. Experiments using the MUSE unlearning benchmark show that SURE can forget sensitive data effectively while also retaining its unlearning performance robustly under quantization.

**Strengths:**

- [S1] **Interesting topic and observation.** The field of LLM unlearning is of growing interest and could be of great potential use to the ML community in practice. The observation that a simple 4-bit quantization of an unlearned model can restore "forgotten" knowledge is very interesting, and would benefit many researchers working on machine unlearning.
- [S2] **Strong empirical results.** Experiments show that the proposed method SURE consistently retains its unlearning efficacy after 4-bit quantization, which nicely demonstrates the paper's hypothesis.

**Weaknesses:**

- [W1] **Narrow scope of the paper.** There exist a number of ways to attack an unlearned model into generating "forgotten" data such as jailbreaking or in-context relearning [A], but the paper and its newly proposed method is solely motivated under one particular attack mechanism (i.e., quantization), which makes the overall scope of the paper rather narrow. To make things worse, the main issue of restoring "forgotten" data does not appear when using 8-bit quantization, but only appears with 4-bit (or less) quantization, which narrows down the motivation even further to only particular scenarios within LLM unlearning.
- [W2] **Weak novelty of saliency-based SURE approach.** While the motivational results of quantization restoring forgotten knowledge is novel, the SURE framework proposed to counteract the restoration is not that novel methodologically, considering that previous work have already explored saliency-based unlearning using the same exact saliency map [B].

[A] Lynch et al., Eight Methods to Evaluate Robust Unlearning in LLMs. arXiv Feb 2024.\
[B] Fan et al., SalUn: Empowering Machine Unlearning via Gradient-based Weight Saliency in Both Image Classification and Generation. ICLR 2024.

**Questions:**

- [Q1] When using parameter-efficient fine-tuning (PEFT) techniques such as LoRA, we tend to use larger learning rates compared to full-finetuning, but its low-rank structure naturally imposes a regularization which could be desirable in mitigating potential bias on small retain sets. Have the authors tested whether the observations on quantizing unlearned models hold under PEFT-based unlearning as well? Any insights would be great.

---

> ### Author Response · Authors · 2024-11-20
>
> We are glad that the reviewer recognizes the interesting problem identified in our paper, and we thank the reviewer for acknowledging our strong empirical results. Below, we provide our point-by-point response to the reviewer's comments:
>
> ---
>
> **W1: narrow scope**
> > There exist a number of ways to attack an unlearned model into generating "forgotten" data
>
> * Our motivation is not to attack the unlearned model and force it to recover the knowledge. Instead, we consider the problem from the perspective that, as quantization becomes increasingly popular in the era of LLMs, it is important to evaluate whether existing unlearning methods remain effective when the unlearned model is quantized. Imagine a data owner requests the removal of copyrighted data from an open-source model, the model provider must apply an unlearning algorithm to update it. The data owner can then test the updated model to verify unlearning. Robust unlearning is crucial to prevent knowledge recovery via quantization, as failure may expose the provider to legal risks for non-compliance
>
> * We thank the reviewer for pointing out the related work [1]. However, according to the results presented in their paper, the performance of forcing an unlearned model to generate the supposed forgotten knowledge is poor. Specifically, the unlearned model retains only 10% of the knowledge, and none of the attack methods achieve the goal of forcing the unlearned model to retain 20% of the knowledge. These results are much less effective compared to our empirical findings. Moreover, the attack methods mentioned in [1], such as Jailbreak and In-context Learning, require significantly more effort to recover knowledge from an unlearned model compared to the simplicity and effectiveness of quantization.
>
> > the main issue of restoring "forgotten" data only appears with 4-bit (or less) quantization
>
> 4-bit quantization is widely studied [2][3], and there is even significant research focusing on lower-bit formats, such as 1-bit quantization [4][5]. Quantization is a widely used technique for deploying LLMs in resource-constrained scenarios, and advanced quantization methods focusing on low-bit quantization are attracting increasing research interest. Our empirical results in Table 1 also demonstrate that the 4-bit model does not exhibit significant utility degradation compared to the full-precision version. Thus, we believe the identified issue of quantization leading to unlearning performance degradation will spark significant research interest within the community.
>
> ---
>
> **W2: novelty of the proposed method**
>
> We acknowledge that localization-informed unlearning methods have been explored in several existing works, and we have appropriately cited the relevant studies in Section 6.1, where we introduced our approach.
> However, we would like to emphasize that the novelty and contribution of the proposed method lie in the deep insights that guided its design. The experimental results provide further insights for future research. The **core contributions of our paper primarily lie in identifying a novel problem, providing a theoretical analysis, and making an initial effort to mitigate this issue while inspiring future research**. Specifically:
>
> * Based on our theoretical analysis, **we identify increasing weight changes as a promising approach that has been overlooked by existing unlearning methods.**. To empirically validate our analysis, we tested a method to enlarge the learning rate, and our results in Table 3 demonstrate that this approach shows promise in preventing knowledge recovery via quantization.
> * However, as demonstrated in the experimental results in the ablation study (Appendix H), a large learning rate can lead to model performance degradation. Thus, we extend the concept of localization-informed unlearning methods to the domain of LLMs by calculating module-level gradients and constructing saliency maps. It is important to note that this is not the only approach to mitigating the negative impact of a large learning rate. **Our aim here is to make an initial effort to empirically verify whether localization-informed unlearning could be a promising solution to help mitigate the effects of a large learning rate**.
>
> Overall, our initial efforts validate the theoretical analysis and inspire future research aimed at increasing weight changes during unlearning to prevent knowledge recovery via quantization. We thank the reviewer for pointing this out and will include a discussion in a future version of our paper.
>
> ---
>
> [1] Eight Methods to Evaluate Robust Unlearning in LLMs. arXiv Feb 2024.
>
> [2] GPTQ: Accurate Post-Training Quantization for Generative Pre-trained Transformers. ICLR 2023.
>
> [3] AWQ: Activation-aware Weight Quantization for On-Device LLM Compression and Acceleration. MLSys 2024.
>
> [4] OneBit: Towards Extremely Low-bit Large Language Models. arXiv Feb 2024.
>
> [5] Billm: Pushing the limit of post-training quantization for llms. arXiv Feb 2024.

---

> ### Author Response · Authors · 2024-11-21
>
> **Q1: PEFT techniques**
>
> Thank you for your insightful question. According to the results in the RWKU benchmark [6], specifically Table 1, using LoRA leads to the unlearned model retaining 87% of the knowledge, which we consider a failure of unlearning, even though the learning rate of LoRA was set to be ten times higher than that for full fine-tuning. Then the results on whether quantization can lead to knowledge recovery may be misleading, as quantization inherently results in a drop in model utility. Thus, our paper focuses on investigating the effect of quantization on unlearning methods that achieve strong unlearning performance in full precision. We thank the reviewer for pointing this out and will include the discussion in a future version of our paper.
>
> [6] RWKU: Benchmarking Real-World Knowledge Unlearning for Large Language Models. NeurIPS D&B Track 2024.

---

> > ### Comment · Reviewer_ZUW5 · 2024-11-25
> >
> > Thank you authors for the detailed response. All of my concerns have been addressed, and I agree that the main contribution lies in discovering an interesting problem for unlearning research: a simple quantization can recover unlearned knowledge. Hence I raise my score to a 6.

---

> > > ### Author Response · Authors · 2024-11-26
> > >
> > > We are glad to hear that the reviewer's concerns have been addressed, and we sincerely thank you for your approval.
> > >
> > > Best regards,
> > >
> > > The Authors

---

> ### Author Response · Authors · 2024-11-25
> **Gentle Reminder**
>
> We sincerely appreciate the insightful comments provided by Reviewer ZUW5. Following the reviewer’s valuable suggestions, we have further clarified the motivation and contributions of our paper. As a friendly reminder, the rebuttal deadline is approaching. We would greatly appreciate it if the reviewer could review our responses at their earliest convenience. We hope that the additional discussions effectively address the reviewer’s concerns and would be deeply grateful if the reviewer considers raising the score.

---

### Official Review · Reviewer_XDZG · 2024-11-04

**Soundness:** 3
**Presentation:** 3
**Contribution:** 3
**Rating:** 8
**Confidence:** 4

**Summary:**

This paper outlines a critical flaw of current unlearning approaches in LLMs - quantizing model weights can sometimes recover “unlearned” information.  The authors attribute this property to the common practice of using low learning rates in unlearning finetuning.  While commonly used to ensure that model capabilities aren’t unlearned due to large gradients, low unlearning rates incentivize minimal weight changes, which means that the quantizations of the weights will be identical before and after unlearning.

They perform extensive experiments to measure if this is the case for several different unlearning loss functions and datasets.  In addition, they propose two simple remedies for this problem - 1) using larger learning rates, and 2) only training components within the transformer that have a high gradient attribution to performance on the forget set.

**Strengths:**

- The paper identifies a critical flaw in current unlearning methodologies and demonstrates a major consequence - lack of robustness to quantization. This calls for closer examination of learning rates used in the unlearning literature.

- The experimental validation is thorough, covering two datasets, multiple metrics for measuring unlearning performance, and different unlearning approaches (gradient ascent and negative preference optimization). This comprehensive evaluation builds confidence that the identified problem is common rather than an aspect of the particular setup being studied in the paper.

- The authors propose a compelling theoretical explanation for why unlearning appears to be so brittle to quantization, involving minimal weight changes due to low learning rates.

**Weaknesses:**

- It is unclear whether robustness to quantization is a more useful way of measuring unlearning robustness compared to few-shot finetuning, which has been studied more extensively recently. The explanation for the mechanism behind vulnerability to quantization - minimal weight changes during unlearning - suggests that both quantization and few-shot finetuning could exploit similar weaknesses.

- Recent unlearning methods [1] [2] [3] potentially encourage deeper forgetting in model representations, leading to larger weight changes. It is unclear whether the findings about quantization vulnerability apply to these newer approaches.

- The SURE method's selective finetuning of model components is similar to recent work that focuses on localizing relevant components for unlearning [4] [5]. The authors should contextualize SURE within this broader research landscape.

[1] Tamirisa et al. (2024). Tamper-Resistant Safeguards for Open-Weight LLMs.

[2] Sheshadri et al. (2024). Latent Adversarial Training Improves Robustness to Persistent Harmful Behaviors in LLMs.

[3] Li et al. (2024). The WMDP Benchmark: Measuring and Reducing Malicious Use With Unlearning.

[4] Guo et al. (2024). Mechanistic Unlearning: Robust Knowledge Unlearning and Editing via Mechanistic Localization.

[5] Bayazit et al. (2024). Discovering Knowledge-Critical Subnetworks in Pretrained Language Models.

**Questions:**

- Have you tested whether the quantization vulnerability patterns hold for models unlearned using these newer approaches?

- Is there any evidence that suggests that models are more vulnerable to quantization than few-shot fine-tuning attacks?

---

> ### Author Response · Authors · 2024-11-23
>
> We thank the reviewer for recognizing the importance and novelty of the problem we identified, as well as the comprehensive experiments presented in our paper. Below, we provide a point-by-point response to the reviewer’s comments:
>
>
> ---
>
>
> **W1 & Q2: whether quantization is a more useful way of measuring unlearning robustness compared to few-shot finetuning**
>
> Thank you for your insightful question.
>
> * Our motivation differs inherently from few-shot fine-tuning. We consider the problem from the perspective that, as quantization becomes increasingly popular, it is crucial to evaluate whether existing unlearning methods remain effective when applied to quantized models.
>
> * According to the results in [1], **the performance of extracting 'supposedly' unlearned knowledge through few-shot fine-tuning is poor**. Specifically, the unlearned model retains only 10% of the knowledge, with 10-shot fine-tuning recovering around 10% of the knowledge and 2-shot fine-tuning recovering only 5%. These results are significantly less effective compared to our empirical findings.
>
> * Moreover, **few-shot fine-tuning requires significantly greater effort to recover knowledge** from an unlearned model compared to the simplicity and effectiveness of quantization.
>
>
> ---
>
>
> **W2 & Q1: experiments on recent unlearning methods**
>
> Thanks for your insightful question. Following your advice, we adopt the unlearning methods RMU [2] and LAT [3] and conduct experiments on the BOOKS dataset. As the idea in [4] is similar to LAT in [3], we only implement LAT here. For LAT, we incorporate it with RMU to form the method RMU_LAT. We tune the learning rate using grid search in [5e-6, 1e-5, 5e-5] to balance unlearning performance and model utility. The results are shown in the table below. It is evident that, for the unlearning metrics M1 and M2, both methods show poor performance after quantization. Our results further highlight the challenge of balancing the prevention of knowledge recovery through quantization with maintaining model utility.
>
>
>
>
> |     Method                 | M1   | M2  | M3   | M4  | Gen  | Tru  | Fac  | Flu   |
> |----------------------|------|-----|------|-----|------|------|------|-------|
> | Target               | 99.8 | 59.4| -57.5| 66.9| 28.7 | 33.6 | 9.1  | 573.3 |
> | RMU                  | 14.2 | 0.0 | -28.3| 0.0 | 29.8 | 36.5 | 0.0  | 69.3  |
> | RMU+4-bit Quan.      | **50.5** | **29.3**| **-57.2**| **41.8**| 25.1 | 32.3 | 4.9  | 689.5 |
> | RMU_LAT              | 11.7 | 0   | -40.5| 0   | 30.9 | 36.3 | 0.0  | 1.99  |
> | RMU_LAT+4-bit Quan.  | **43.0** | **27.4**| **-57.6**| **37.9**| 25.7 | 36.9 | 4.8  | 685.9 |
>
>
>
>
> ---
>
>
> **W3: further contextualize proposed method with related work**
>
> We acknowledge that localization-informed unlearning methods have been explored in several existing works, and we appropriately cited the relevant studies in Section 6.1, where we introduced our approach.
>
> * Based on our theoretical analysis, **we identify increasing weight changes as a promising approach that has been overlooked by existing unlearning methods.**. To empirically validate our analysis, we tested a method to enlarge the learning rate, and our results in Table 3 demonstrate that this approach shows promise in preventing knowledge recovery via quantization.
> * However, as demonstrated in Section 6.1 and through the experimental results in the ablation study (Appendix H), a large learning rate can lead to model performance degradation. Thus, we extend the concept of localization-informed unlearning methods to the domain of LLMs by calculating module-level gradients and constructing saliency maps. It is important to note that this is not the only approach to mitigating the negative impact of a large learning rate. **Our aim here is to make an initial effort to empirically verify whether localization-informed unlearning could be a promising solution to help mitigate the effects of a large learning rate**.
>
> We thank the reviewer for pointing this out and will include a discussion in a future version of our paper.
>
>
>
> ---
>
>
> [1] Eight Methods to Evaluate Robust Unlearning in LLMs. arXiv Feb 2024.
>
> [2] Sheshadri et al. (2024). Latent Adversarial Training Improves Robustness to Persistent Harmful Behaviors in LLMs.
>
> [3] Li et al. (2024). The WMDP Benchmark: Measuring and Reducing Malicious Use With Unlearning.
>
> [4] Tamirisa et al. (2024). Tamper-Resistant Safeguards for Open-Weight LLMs.

---

> > ### Comment · Reviewer_XDZG · 2024-11-26
> >
> > Thanks, these additional experiments are very useful.  I will increase my score.

---

> > > ### Author Response · Authors · 2024-11-26
> > >
> > > We sincerely thank the reviewer for dedicating time to reviewing our paper and providing insightful comments. We greatly appreciate your approval of our work.
> > >
> > > Best regards,
> > >
> > > The Authors

---

### Official Review · Reviewer_4v3P · 2024-11-04

**Soundness:** 2
**Presentation:** 2
**Contribution:** 3
**Rating:** 6
**Confidence:** 4

**Summary:**

This paper addresses the problems that arise when quantizing models after unlearning. The authors also propose a quatization-robust unlearning algorithm.

**Strengths:**

1. The paper is original to my knowledge.
2. The writing is clear and flows well.
3. The problem is well motivated --- I do think unlearning is largely studied in full precision and the effects of quantization are important to understand given that its a common practice.

**Weaknesses:**

1. **Presentation** I found the results incredibly difficult to parse. The large tables are hard to exact trends from and I think at times the results that matter are actually split across more than one of these tables or duplicated. For example, Table 3 has some of the data in Table 1 but it's missing other rows one might care about. I was flipping back and forth a bunch and it is very hard to internalize the trends. An informative visualization of this data is missing.
2. **Unclear Experimental Setup** I see in the formal notation that $f_\text{target}$ is the base model, as in one that should know the sensitive/private info and one one which unlearning algorithms haven't yet been run. It isn't clear what that model is -- what size, what training data, how it is trained etc.
3. **Limited Data Scope** The experiments make use of two datasets that I have not seen in many other unlearning papers.

**Questions:**

1. **Presentation** Can you provide some compelling plots here? Is the point that the pink rows in Table 1 are bad? The have still unlearned, just not as much as the full precision. Can you elaborate on the problem here and on how big an effect the solution offers? Overall, I'm hoping you can convey the issue and the positive results in plots or otherwise in concise tables. The results are very hard to parse.
2. **Unclear Experimental Setup** What is $f_\text{target}$? What has it been trained on? Do all your experiments consider the same model? How do things change with other models? Larger models?
3. **Limited Data** What is in the Books and the News datasets that might need to be unlearned? What can we learn from these experiments that might translate to private data or sensitive data? Can the experiments be expanded to use common unlearning tests (TOFU (for individual privacy), WMDP (for toxic output), HarryPotter etc.)?

---

Post rebuttal I moved my score up to advocate that this paper be accepted.

---

> ### Author Response · Authors · 2024-11-21
>
> We thank the reviewer for recognizing the importance and novelty of the problem identified in our paper. Below, we provide a point-by-point response to the reviewer's comments:
>
> **W1 & Q1: presentation**
> >  Table 3 has some of the data in Table 1 but it's missing other rows one might care about.
>
> As demonstrated in our experimental analysis in Section 4.4, Table 3 focuses on showing that unlearned models with 4-bit quantization, using techniques GPTQ and AWQ, exhibit poor unlearning performance compared to their full-precision counterparts. To maintain simplicity, we removed redundant rows from Table 1. Meanwhile, we retain the results for $f_{target}$, its 4-bit quantized version, and $f_{retain}$ to enable readers to easily compare the unlearning performance among these models.
>
> > Is the point that the pink rows in Table 1 are bad? 4-bit model still unlearned, just not as much as the full precision
>
> In Table 1, the pink rows represent the results for the unlearned model with 4-bit quantization using the round-to-nearest (RTN) technique.
> As the reviewer observed, an unlearned model with 4-bit quantization still unlearns to some extent. However, compared to the full-precision version, its performance is significantly poorer. This observation encapsulates the core issue we identified in this paper.
>
> > Can you provide some compelling plots here?
>
> Due to time constraints during the rebuttal, we will provide plots to visualize the results in a future version of our paper. However, in our initial submission, **we provided a detailed analysis of the experimental results in Sections 4.2, 4.3, 4.4, and Appendix E**.
>
>
>
>
>
> ---
>
>
>
> **W2 & Q2: Experimental setup**
> > Details of $f_{target}$, including its size, training data, and training process.
>
> * Thanks for pointing this out. We have included additional experimental details in the Appendix C.3 of the updated version of our paper, including the details of $f_{target}$ and $f_{retain}$.
>
> * Following the experimental setting in MUSE, for NEWS dataset, the backbone model is LLama2-7B [1], for BOOKS dataset, the backbone model is ICLM-7B [2] which share the same architecture with LLama2-7B but does not contain the Harry Potter books in its pretraining data.
>
>
> ---
>
>
> **W3 & Q3: Data scope**
> > What is in the Books and the News datasets that might need to be unlearned?
>
> In our main pages, we conduct experiments on the MUSE benchmark and follow its settings, using two datasets: NEWS and BOOKS. The NEWS dataset consists of BBC news articles collected after August 2023. The BOOKS dataset comprises the Harry Potter book series.
>
> > What can we learn from these experiments that might translate to private data or sensitive data?
>
> In Appendix F, we conduct experiments on the RWKU [3] benchmark, which includes private data to be unlearned. The data source for this benchmark consists of a list of famous individuals scraped from The Most Famous All-Time People Rank. These entities are linked to Wikipedia, and their popularity is measured using Wikipedia page views. The top 200 most popular entities, based on page view rankings, are selected as unlearning targets. Our empirical results consistently show that quantization leads to poor performance in unlearning.
>
> > Can the experiments be expanded to use common unlearning tests (TOFU (for individual privacy), WMDP (for toxic output), HarryPotter etc.)?
>
> For TOFU (individual privacy), we present empirical results on RWKU, which involves similar content to be unlearned.
> For WMDP, running their benchmark requires data that must be requested; unfortunately, we were unable to obtain the data during the rebuttal period.
> For Harry Potter, our empirical results on the BOOKS dataset include this content. **Overall, our empirical results comprehensively cover three domains of knowledge sources: news articles, copyrighted book content, and individual private information.**
>
> Furthermore, our empirical results and theoretical analysis are agnostic to the content of the forget dataset. This means that the specific text to be unlearned typically has no impact on the results.
>
>
>
> ---
>
>
>
> [1] Llama 2: Open Foundation and Fine-Tuned Chat Models. arXiv.
>
> [2] In-context pretraining: Language modeling beyond document boundaries. ICLR 2024.
>
> [3] RWKU: Benchmarking Real-World Knowledge Unlearning for Large Language Models. NeurIPS D&B Track 2024.

---

> > ### Comment · Reviewer_4v3P · 2024-11-21
> > **Reviewer response and score update**
> >
> > Thank you for addressing my points. In light of the rebuttal I will increase my score.
> >
> > I have some closing thoughts about the final draft that authors should make an effort to incorporate to make a stronger presentation of their work:
> > 1. I do think the final draft of the paper should have the details of the model in the body not the appendix.
> > 2. The choice of MUSE and the fact (in the authors words) that "Overall, our empirical results comprehensively cover three domains of knowledge sources: news articles, copyrighted book content, and individual private information." need to be better addressed/motivated/discussed. While my initial impression may not have been perfect, it should serve as an example of what other unlearning researchers might take away from the paper.
> > 3. I do not think that missing plots is grounds for rejection (and I advocate we accept this paper) but I will strongly suggest to the authors that the current presentation of the results be revised. The main points are difficult to extract from the tables as they are. Consider plots or refinements to the tables.
> > 4. I agree with the other reviewers on several of their points---please make more clear why quantization is a problem if big corps who serve their models on servers don't need to do this. In other words, give some justification that there is a risk here, openly distributed weights that individuals may quantize for local inference is not a practical scenario as the initial (pre unlearning) weights would already be out there in the public so sharing the post-unlearning weights holds little risk even if downstream users quantize. Please mention this and further motivate your work in the final draft.
> > 5. Another point on which I agree with other reviewers is in non-finetuning based methods. I see in your response to them that these methods don't perform well on some of your tasks -- include this point and mention in the final version that other unlearning techniques exist and may not be vulnerable to quantization problems, but still offer less unlearning performance across the board.
> >
> > Thank you for your time, and please don't feel obligated to respond to this message if you are pressed for time.

---

> > > ### Author Response · Authors · 2024-11-24
> > >
> > > We sincerely thank the reviewer for recognizing our efforts and increasing the score. We appreciate your valuable feedback and will include the discussion in our final version.
> > >
> > > Best regards,
> > >
> > > The Authors

---

### Author Response · Authors · 2024-12-04
**Summary of Manuscript Revision**

We sincerely thank all the reviewers for reviewing our work and providing thoughtful feedback. **We are grateful that all reviewers agree we have identified an important, novel, and interesting issue: the quantization of an unlearned model can result in knowledge recovery.**

In response to the reviewers' comments and suggestions, we have implemented the following updates in the newly revised version:

* In Line 264 and Appendix C.3, we provide details of the retrained model and the target model.
* In Appendix I, we include additional results and discussions on other unlearning methods.
* In Appendix J, we offer further insights into the core contributions of our paper.
* In Appendix K, we highlight the critical nature and real-world scenarios of the identified problem where the quantization of an unlearned model can restore "forgotten" knowledge.
* In Appendix L, we compare quantization with other methods for evaluating robust unlearning in LLMs.
* Other minor revisions, including adjustments to the width of Table 4 and font corrections.

---

### Meta-Review · Area_Chair_gc6R · 2024-12-20

**Metareview:**

This paper examines LLMs after unlearning, a process that removes certain knowledge from the model. The authors identify a critical failure mode: when the weights of an unlearned LLM are quantized, the supposedly removed knowledge can be recovered. They then propose a saliency-based parameter gradient clipping method (SURE) to mitigate the problem.

The reviewers’ comments highlight several strengths of the paper. They agree on the importance of the observation about quantization undermining unlearning. They also acknowledge its clear motivation (Reviewers 4v3P, ZUW5, KTWe), sufficient empirical experiments (Reviewers XDZG, ZUW5), and theoretical analysis of the identified issue (Reviewers 4v3P, Sbbj).

However, the reviewers also express concerns, including unclear experimental setup (Reviewer 4v3P), misleading presentation (Reviewers 4v3P, KTWe), limited evaluation of unlearning methods (Reviewers XDZG, Sbbj), limited novelty of the SURE method (Reviewers XDZG, ZUW5, Sbbj).

During the rebuttal, the authors made great efforts and successfully addressed most of the reviewers’ concerns. Considering all factors, the AC agrees with reviewers on the significance of finding quantization undermines unlearning and therefore recommends acceptance of this paper. Nonetheless, the AC suggests that the authors include additional experiments and discussions during rebuttal in the final version to enhance the clarity of this paper.

**Additional Comments On Reviewer Discussion:**

The authors made great efforts to address the reviewers' concerns in their rebuttal. After the rebuttal discussion, most issues raised by the reviewers were resolved. Reviewers 4v3P, XDZG, and ZUWS increased their initial ratings, and most reviewers agreed to accept the paper.

However, one reviewer (Sbbj) remained concerned that quantization alone cannot recover knowledge of a collapsed model after unlearning training. The authors provided detailed responses and additional experiments to address this concern, but reviewer Sbbj did not give feedback on the new response, possibly due to time constraints.

Taking all factors into account, AC recommends accepting the paper.

---

### Decision · Program_Chairs · 2025-01-22

Accept (Poster)